# Decoupled oxidation process enabled by atomically dispersed copper electrodes for in-situ chemical water treatment

Ziwei Yu[1], Xuming Jin[1], Yang Guo[1], Qian Liu[2], Wenyu Xiang[1], Shuai Zhou[1], Jiaying Wang[1], Dailin Yang[1], Hao Bin Wu ●[2] ✉ & Juan Wang ●[1] ✉

In-situ wastewater treatment has gained popularity due to cost and energy savings tailored to water sources and user needs. However, this treatment, particularly through advanced oxidation processes (AOPs), poses ecological risks due to the need for strong oxidizing agents. Here, we present a decoupled oxidation process (DOP) using single-atom copper-modified graphite felt electrodes. This process creates a positive potential difference (ΔE ~ 0.5 V) between spatially isolated oxidants and organics and drives electron transfer-based redox reactions. The approach avoids the drawbacks of conventional AOPs, while being capable of treating various recalcitrant electron-rich organics. A floating water treatment device designed based on the DOP approach can degrade organic molecules in large bodies of water with oxidants stored separately in the device. We demonstrate that over 200 L of contaminated water can be treated with a floating device containing only 40 mL of oxidant (10 mM peroxysulphate). The modular device can be used in tandem structures on demand, maximizing water remediation per unit area. Our result provides a promising, eco-friendly method for in-situ water treatment that is unattainable with existing techniques.

Anthropogenic wastewater production has led to significant concerns on water contamination worldwide[1-5]. In modern society, centralized water treatment strategies have been adopted for the most of wastewater treatment, by which wastewater is collected in sewerage systems and treated in water treatment plants[4,6]. Despite their usefulness, there are difficulties with the construction and maintenance of centralized infrastructures, especially for geographically isolated communities and open surface streams such as rivers, ponds and wetlands[7,8]. Thus, it is an urgent need to develop in-situ techniques to remediate polluted water on site and immediately restore the ecological environment[9-11]. Biodegradation of water contaminants by living organisms (e.g., bacteria, fungi and microfauna) is by far one of the most commonly used techniques for in-situ water remediation[12-16]. With the emergence of new bio-refractory organics (polycyclic aromatics, chlorinated hydrocarbons, etc.) in water, there is an increasingly high demand for the development of environmentally friendly in-situ strategies for dealing with recalcitrant organic pollutants in water.

Chemical water treatment with oxidant-involved advanced oxidation processes has shown promise in efficiently mineralizing various recalcitrant organic contaminants[5,17-21]. These AOPs generally involve catalytic decomposition of strong oxidants (e.g. $O_3$, $H_2O_2$ and persulfate) to produce reactive oxygen species (ROS; e.g. •OH, $SO_4^{•-}$ and $^1O_2$)[19,20,22,23]. The as-produced ROS then facilitate oxidative degradation of organic pollutants in water due to their relatively high redox potentials (e.g., $SO_4^{•-}$, 2.5–3.1 V vs. NHE; •OH, 1.9–2.7 V vs. NHE)[19,20,22]. However, the introduction of strong oxidants and ROS to water may lead to indiscriminate damage to

[1]College of Environmental & Resource Sciences, Zhejiang Provincial Key Laboratory of Organic Pollution Process and Control, Zhejiang University, Hangzhou 310058, China. [2]Institute for Composites Science Innovation (InCSI) and State Key Laboratory of Silicon Materials, School of Materials Science and Engineering, Zhejiang University, Hangzhou 310027, China. ✉e-mail: hbwu@zju.edu.cn; wjuan@zju.edu.cn

aquatic organisms and ecosystems[24,25]. Moreover, the catalysts and resulting by-products (e.g., $Fe^{3+}$, $SO_4^{2-}$, etc.) pose potential hazardous, leading to secondary pollution concerns[26–28]. Despite recent focus on less environmentally vulnerable methods such as non-radical AOPs[29–31], achieving hazard-free, in-situ water treatment remains a major challenge, especially for surface streams with complex aquatic ecosystems (Fig. 1).

Here, we propose a decoupled oxidation process to realize an environmentally friendly chemical water treatment. This approach relies on efficient catalyst capable of creating a sufficient positive potential difference (ΔE) between the two oxidative and reductive half-reactions within an AOP, thus triggering the two decoupled reactions. Single-atom catalysts have emerged as a promising material design option for various catalytic processes, as the efficiency of atomic reaction sites can be maximized to nearly 100%[32–35]. We identified a copper-based single-atom catalyst (Cu-N-C) with high catalytic activity to create a substantial potential difference (ΔE = 0.5 V) between the oxidative and reductive two half-reactions. The high loading of the composite with atomically dispersed Cu atom sites (16.37 wt%) provides abundant reaction sites (adsorption and catalytic sites) to facilitate catalytic reactions. Consequently, we can conduct the oxidation of organic pollutants in contaminated water while relegating the reduction of oxidants to an isolated floating chamber. This strategy can also be validated for other commercial catalysts such as CuO and carbon nanotubes. Thus, we provide a general strategy to prevent direct contact between the aquatic environment and oxidative additives, thereby eliminating potential environmental hazards associated with AOP (Fig. 1).

## Results

### Setup and performance evaluation of the DOP

In a typical experiment, we employed a double-chamber galvanic setup with Cu single atoms/nitrogen-doped porous carbon-modified graphite felt (Cu-N-C@GF) electrodes to demonstrate the proposed DOP approach (Fig. 2a, b and Supplementary Fig. 1). In the experimental setup, a Nafion proton exchange membrane (PEM) was applied to block the exchange of solvents/reactants while allowing the migration of small cations (e.g. $H^+$, $Na^+$ ions) between the two chambers to maintain charge neutrality during reactions. Bisphenol A (BPA) was selected as a model organic contaminant, since it is a biorefractory endocrine disruptor widely detected in polluted water[36–38]. Peroxysulphate (PDS) was used as the oxidant for the DOP, considering the relatively low cost and high stability of PDS compared with other oxidants such as peroxymonosulfate (PMS) and $H_2O_2$[39].

Cu-N-C@GF electrode was made by coating the graphite felt (GF) with fabricated Cu single atoms/nitrogen-doped porous carbon (Cu-N-C) composites (Fig. 2c and Supplementary Figs. 2–7). The Cu-N-C was synthesized by calcination of pre-fabricated $Cu(BTC)(H_2O)_3$ with dicyandiamide (DCD) at 800 °C for five hours, under argon atmosphere (Supplementary Figs. 2 and 3)[40]. Low-resolution transmission electron microscopy (LR-TEM) showed no formation of crystallized particles (Supplementary Fig. 4a).The as-prepared composite has a porous structure with relatively large surface area (~200 m² g⁻¹), which provides sufficient active sites for reactions (Supplementary Fig. 4b–e and Supplementary Table 2). The X-ray diffraction (XRD) pattern of Cu-N-C only exhibited peaks related to the graphitic carbon structure, while the elemental mapping images unveiled the uniform dispersion of N, C and Cu elements within the whole composites (Supplementary Fig. 4c, g).

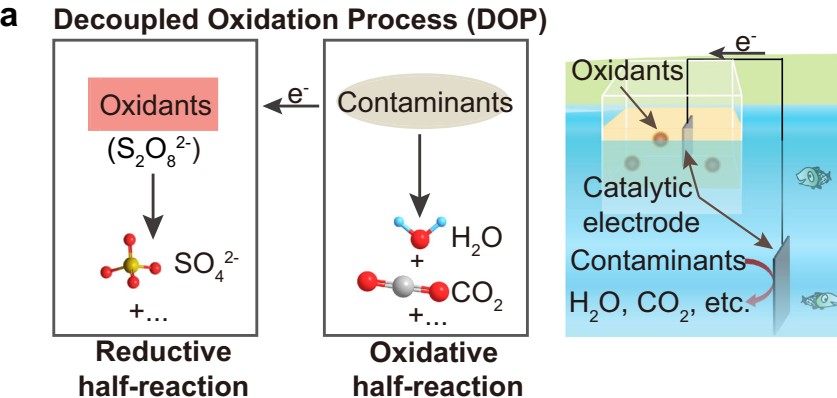

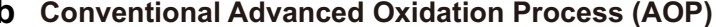

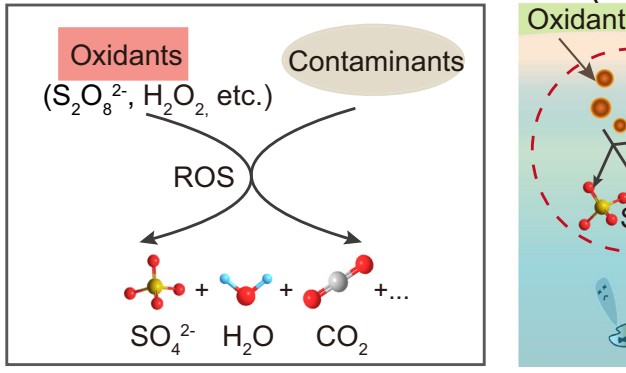

**Fig. 1 | Comparison between a decoupled oxidation process (DOP) and a conventional advanced oxidation process (AOP). a** Schematically shows the DOP method which spatially separates oxidant reduction and organic contaminant oxidation, preventing direct contact between harmful oxidants and contaminated water. **b** Illustrates conventional AOPs, where oxidant chemicals are directly added to contaminated water, potentially posing hazards to the environment and leading to the formation of secondary pollutants.

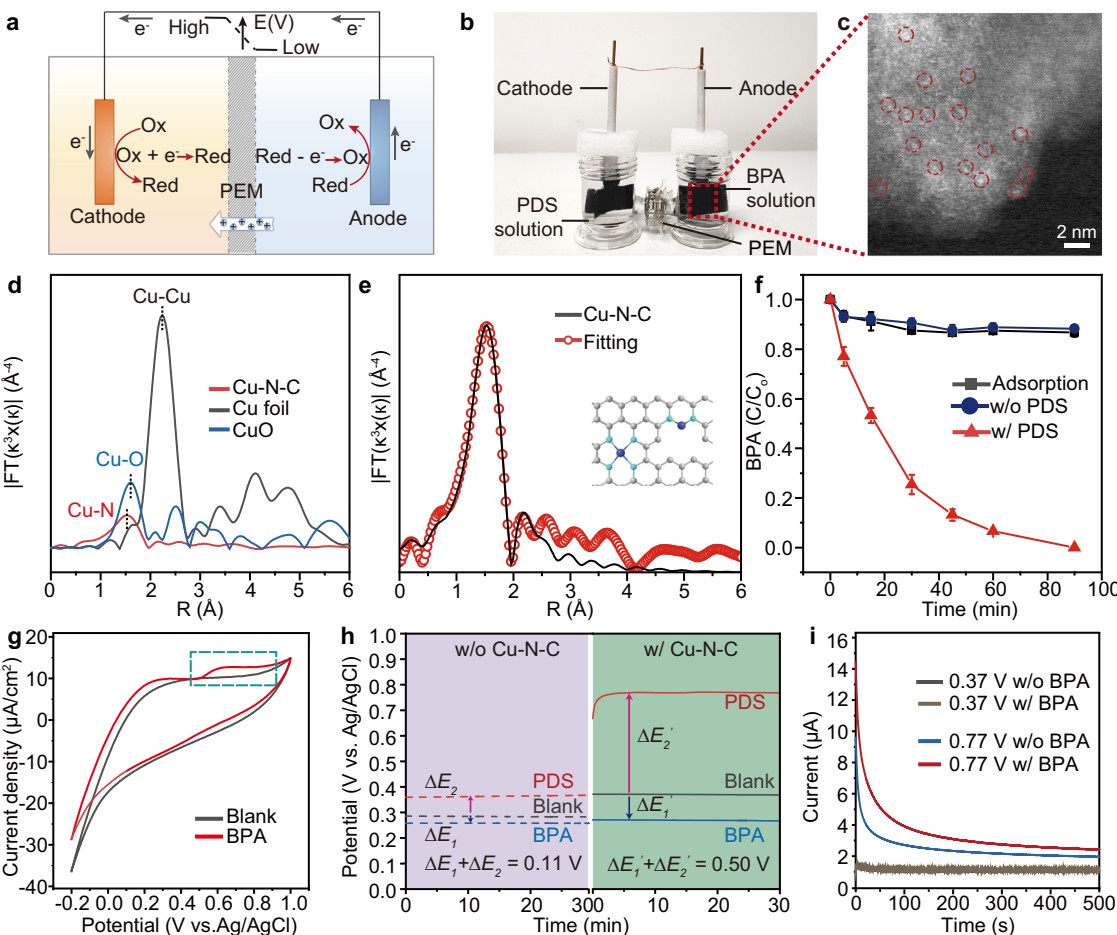

**Fig. 2 | Design and performance evaluation of the DOP. a** Schematic diagram of the DOP conducted in a double-chamber reactor, with an interconnection part sealed with a proton exchange membrane (PEM). Ox refers to oxidants and Red refers to reductants. **b** Photograph displaying the double-chamber reactor utilized for testing the DOP's performance. **c** HADDF-STEM image of the synthesized Cu-N-C composite coated on the graphite felt electrode. **d** EXAFS spectra of Cu-N-C at the Cu K-edge compared with reference samples. **e** Corresponding FT-EXAFS fitting analysis of Cu-N-C. **f** Monitoring of the BPA concentration changes in the anodic chamber under different conditions. Reaction conditions: BPA, 5 µM; PDS, 10 mM; NaCl, 0.5 wt‰. Experiments were conducted in triplicate, and the error bars represent the arithmetic mean ± standard deviation. **g** Cyclic voltammetry curve measurement on the Cu-N-C@GF electrode with and without the presence of BPA. Conditions: BPA, 0.1 mM; borate buffer solution, 40 mM. **h** Respective open-circuit potential curve measurement of the BPA and PDS solutions, the working electrode with and without the presence of Cu-N-C electrodes. Reaction conditions (if required): BPA, 0.1 mM; PDS, 10 mM; borate buffer solution, 40 mM. **i** Monitoring of current-time curves under different set potentials on the Cu-N-C@GF electrode. Reaction conditions (if required): BPA, 0.1 mM; borate buffer solution, 40 mM.

High annular dark field image-scanning transmission electron microscope (HADDF-STEM) imaging reveals the existence of atomically dispersed Cu on the nitrogen-doped carbon matrix (Fig. 2c). These results confirm that Cu is atomically doped in the matrix. The average oxidation valence of Cu was determined to be between $Cu^0$ and $Cu^{2+}$ by both X-ray absorption fine structure (XAFS) and X-ray photoluminescence spectroscopy (XPS) measurements (Supplementary Fig. 4h and i)[35]. The extended X-ray absorption fine structure (EXAFS) analysis unveiled the presence of only one Cu-N bond in the composite (Fig. 2d)[40,41]. The fitted Fourier transforms of the Cu K edge EXAFS spectra of Cu-N-C identified the coexistence of $Cu-N_2$ and $Cu-N_4$ sites in the Cu-N-C (Fig. 2e and Supplementary Table 3)[40,42]. Due to the high nitrogen content, the Cu-N-C composite exhibits a relatively large Cu loading concentration of 16.37 wt% (Supplementary Fig. 4f).

The ability of the device to remove BPA under a series of conditions was investigated. Once the PDS solution was added to the cathodic chamber, the concentration of BPA in the anodic chamber decreased spontaneously (Fig. 2f). The BPA removal rate reached 100% within 1.5 hours. To unveil whether BPA was physically absorbed on the electrode surface instead of being degraded, we measured cyclic voltammetry (CV) curves. The detection of an oxidation peak in the measured CV curve supports the possibility of electrocatalytic oxidation of BPA on Cu-N-C@GF (Fig. 2g). Following a 2-hour reaction, the chemical oxygen demand (COD) of the added BPA solution decreased to approximately 20% of the initial level (Supplementary Fig. 8a). The degradation products of BPA were identified via ultrahigh performance liquid chromatography tandem four-pole time-of-flight mass spectrometry (UPLC-QTOF-MS/MS), and their toxicity was assessed using the ecological structure-activity relationship model (ECOSAR) to evaluate risks to fish, Daphnia, and green algae (Supplementary Fig. 9). Notably, both acute and chronic toxicity (ChV) of the oxidized products decreased significantly during the degradation process, indicating the efficiency of our system in reducing the toxicity of organic contaminants.

## Mechanistic investigation

To unveil the reaction mechanism, we first performed the corresponding electron paramagnetic resonance (EPR) analysis and scavenger experiments (Supplementary Fig. 10a and b). The results showed no generation of ROS (including •OH, $SO_4^-$ and $^1O_2$)[43,44]. This suggests the degradation of BPA by the experimental set-up did not follow conventional AOPs. We speculated that the BPA removal was attributed to the non-ROS involved redox reaction between the PDS/Cu-N-C and BPA/Cu-N-C pairs. We proposed that the full redox

reaction between PDS and BPA in our system was decoupled into two half-reactions, as shown in Eqs. 1 to 3.

Full redox reaction:

$$S_2O_8^{2-} + BPA \rightarrow SO_4^{2-} + oxBPA \qquad (1)$$

(Note: oxBPA refers to the oxidative products of BPA)
Cathodic half reaction:

$$S_2O_8^{2-} + 2e^- \rightarrow SO_4^{2-} \qquad (2)$$

Anodic half reaction:

$$BPA \rightarrow oxBPA + 2e^- \qquad (3)$$

To unveil the electrical driving force of these reactions, we implemented chronopotentiometry to monitor the potential changes of Cu-N-C@GF electrodes after adding either PDS or BPA. The addition of PDS or BPA caused a slight potential change at the bare GF electrode (+0.37 V vs. +0.28 V, $\Delta E_o = \sim 0.11$ V, Ag/AgCl as the reference electrode, Fig. 2h). The small potential difference ($\Delta E_o$) between the cathodic PDS/GF and anodic BPA/GF electrodes cannot drive an obvious redox reaction, which is possible due to the unavoidable energy compensation requirement during operation (e.g. electrical resistance /overpotentials)[45,46]. By contrary, the potential of Cu-N-C@GF electrode immediately raised to +0.77 V after adding PDS, while it inversely dropped to +0.27 V upon the addition of BPA. Thus, the employment of Cu-N-C@GF electrodes increased the potential difference between PDS/Cu-N-C@GF and BPA/Cu-N-C@GF electrodes to -0.50 V. It is speculated that the elevated positive potential difference drives the transfer of electrons from BPA to PDS through the Cu-N-C@GF electrodes.

To further manifest this, we monitored the current flow change in the anodic chamber with the Cu-N-C@GF electrode at different stimulated potentials (Fig. 2i). There was no appreciable difference in current flow with/without the presence of BPA at the equilibrium potential of Cu-N-C@GF electrode without PDS (+0.37 V) as the potential is too low to drive electron transfer. When the stimulated potential reached the equilibrium potential of PDS/Cu-N-C@GF (+0.77 V), the current with BPA became much higher than that without BPA, suggesting electron transfer and electro-oxidation of BPA. We argued that the potential difference generating between the cathodic and anodic chambers was the driving force in triggering the redox reaction.

In our investigation, we observed that unmodified graphite felt (GF) electrodes exhibited negligible efficacy in the removal of BPA (Supplementary Fig. 11). We attributed the remarkable catalytic removal of BPA to the coated Cu-N-C. To elucidate the role of Cu-N-C, we introduced the composite directly to a solution mixture of BPA and PDS (Supplementary Fig. 12). Notably, the presence of both Cu-N-C and PDS led to a rapid reduction in BPA concentration (Supplementary Fig. 12a). Scavenger tests and electron paramagnetic resonance (EPR) analysis experiments were also conducted to identify the active species responsible for this catalytic removal, and both showed the absence of common active species from PDS in the solution (including •OH, $SO_4^{•-}$ and $^1O_2$) (Supplementary Fig. 12b, c). We employed chronopotentiometry (Supplementary Fig. 12d) to assess the open-circuit potentials of the electrode. While standalone PDS and bisphenol A systems exhibited no significant potential difference, the introduction of Cu-N-C generated a substantial potential gap ($\Delta E = 0.9 - 0.34 = 0.56$ V) between the Cu-N-C + PDS and Cu-N-C + BPA configurations. This observation suggests that when BPA and PDS encounter Cu-N-C, they adsorb onto its surface, resulting in a positive $\Delta E$ between the adsorbed PDS* and BPA*. This positive $\Delta E$ drives electron transfer from BPA* to PDS*, facilitating oxidation of BPA (Supplementary Fig. 12e). Even when they are physically separated but electrically connected, this potential difference persists and enables electron transfer from BPA to PDS, thus supporting our decoupled oxidation process.

It is widely reported that the calcination temperature always influences the environment of atomically dispersed metals and the functionality. We calcinated the prepared precursor mixture at a temperature of 600 °C, 700 °C and 900 °C, respectively, and named the obtained composites as Cu-N-C-6, Cu-N-C-7 and Cu-N-C-9, accordingly. Since the catalyst Cu-N-C in our design was calcined at 800 °C, it was referred to here as Cu-N-C-8 for ease of understanding. Based on a set of experiments (characterization and analysis of BPA removal performance in Supplementary Fig. 14), we identified that Cu-N-C-8 in our experiment showed the best catalytic performance, as it had the largest exposed surface area as well as the highest loading density of atomically dispersed Cu sites (16.37 wt%). Thus, Cu-N-C-8 was adopted in our further experiments and donoted as Cu-N-C.

## Identification of active sites on the Cu-N-C@GF electrode

To elucidate the active sites on the Cu-N-C@GF electrode for the decoupled oxidation process, we added the chelating agent KSCN and EDTA-2Na. Both resulted in a significantly decreased BPA removal rate (Supplementary Fig. 15), suggesting that the metal sites may be mainly responsible for the reaction. We further conducted density functional theory (DFT) calculations to reveal the interactions of reactants (PDS and BPA) with the functional sites on Cu-N-C. We compared the adsorption energy of PDS and BPA at different sites on Cu-N-C including graphitic C, pyridinic N, CuN$_2$ and CuN$_4$ (Fig. 3a). The result suggests that the CuN$_2$ site has much-higher adsorption capacity for both PDS and BPA compared with the other sites. Additionally, the O-O bond length ($l_{o-o}$) of adsorbed PDS on CuN$_2$ was lengthened to 1.481 Å, while CuN$_4$ showed weaker elongation ($l_{o-o} = 1.479$ Å) compared with the original PDS ($l_{o-o} = 1.426$ Å) (Supplementary Fig. 16)[47]. These results imply that CuN$_2$ and CuN$_4$ on Cu-N-C functionalize as the main sites for the absorption and activation of PDS. We then calculated the charge density differences between the corresponding paired PDS/BPA with the functional sites[48,49]. Upon respective pairing with PDS and BPA (Fig. 3b and c), CuN$_2$ shows the strongest capability to extract electrons from BPA, while it also has considerable ability to donate electrons to PDS. Taken together, we can conclude that CuN$_2$ sites serve as the main reaction sites for carrying out the redox reactions (Fig. 3d).

## Feasibility and versatility of the DOP strategy

Subsequently, we assessed the DOP's applicability to various organic contaminants. It showed successful removal of phenol, p-chlorophenol (p-CP), and 2,4-dichlorophenol (2,4-DCP), except for p-nitrophenol (p-NP) (Fig. 3e). We attributed the specified reaction selectivity to the potential differences between molecules and electrodes of the reaction system. The potential differences ($\Delta E_1$) between the Cu-N-C@GF electrode electrode paired with phenol/p-CP/2,4-DCP were measured to be larger than 0.03 V compared with the bare electrode (Supplementary Fig. 18). In contrast, for p-NP, the $\Delta E_1$ is only around 0.01 V, which is perhaps too small to drive the electron transfer from the organic compound to PDS (Fig. 3f and Supplementary Fig. 18j–l)[45,46]. Thus, it becomes evident that creating a substantial positive $\Delta E$ between the cathode and anode ends is crucial for triggering the redox reaction. In addition to the phenolic compounds, our system also shows obvious treatment efficiency towards other organics, including sulfamethoxazole and furfuryl alcohol (Supplementary Fig. 19). These results suggests the feasibility but also the target selectivity of our system. The effects of common inorganic ions present in natural surface water, natural organic matter (NOM), and pH fluctuations on system performance was systematically investigated. Our system consistently remained at approximate 100% across all these factors, as well as during a ten-time cycled oxidization of BPA, demonstrating the robustness of our system (Supplementary Figs. 20 and 21).

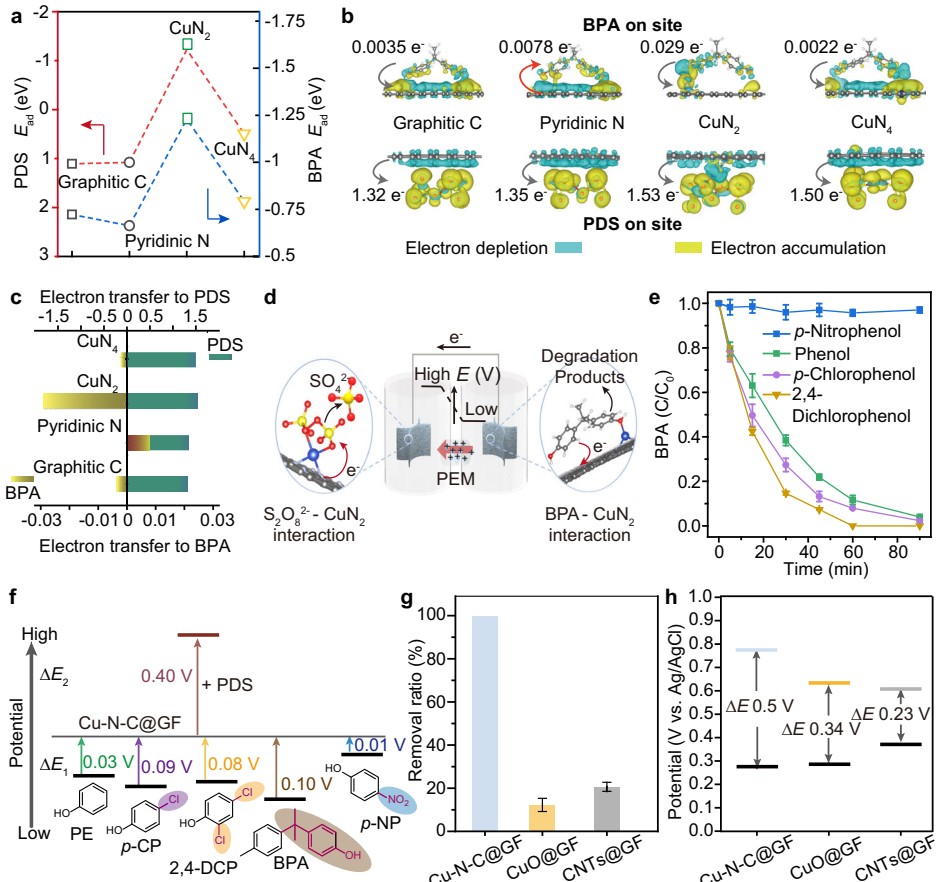

**Fig. 3 | Understanding reaction sites and mechanisms in the DOP. a** Calculated adsorption energy ($E_{ad}$) of different sites on the catalyst surface towards PDS and BPA. **b**, **c** Calculated charge density difference for PDS and BPA absorption, respectively, on different sites on the Cu-N-C surface. Yellow and blue isosurfaces represent charge accumulation and depletion in the space, respectively. **d** Schematic of the possible detailed reaction process facilitated by the $CuN_2$ sites on Cu-N-C@GF electrodes during the DOP. **e** Monitoring of removal performance towards different organic pollutants using the double-chamber reactor. Reaction conditions: PDS, 10 mM; pollutants, 5 μM; NaCl, 0.5‰. Experiments were conducted in triplicate, and the error bars represent the arithmetic mean ± standard deviation. **f** Schematic of the potential differences ($\Delta E_1$, $\Delta E_2$) generated between the electrode Cu-N-C@GF with the respective co-existence of organics/ PDS. **g** Comparison of the BPA removal ratio in the DOP system with different catalytic electrodes: Cu-N-C@GF, CuO@GF and CNTs@GF. Conditions: BPA, 5 μM; PDS, 10 mM; NaCl, 0.5‰. Experiments were conducted in triplicate, and the error bars represent the arithmetic mean ± standard deviation. **h** Comparison of the measured potential differences between the two ends, under different electrode-based double-chamber systems.

To ascertain the broad applicability of our approach, we extended our investigation to the fabrication of graphite felt-based electrodes coated with CuO and carbon nanotubes (CNTs) (Supplementary Figs. 22 and 23). Subsequently, by employing these CuO@GF and CNTs@GF electrodes in the decoupled oxidation reactions, we observed a notable BPA removal capacity in both cases (Fig. 3g, Supplementary Figs. 22e and 23e). Chronopotentiometry analysis of the respective reactions revealed a $\Delta E_{CuO}$ of approximately 0.34 V and a $\Delta E_{CNTs}$ of ~ 0.23 V within the two reaction systems (Fig. 3h, Supplementary Figs. 22f and 23f). This experiment underscores the versatility and generality of our DOP strategy. It is noteworthy that while the CNTs@GF electrode yields larger $\Delta E_{CNTs}$ compared to $\Delta E_{CuO}$, the CNTs@GF electrode system demonstrates better BPA removal efficacy compared to CuO. This disparity can be ascribed to the greater number of reaction sites on the CNTs@GF electrode relative to the CuO@GF counterpart. Consequently, our Cu-N-C@GF-based system not only exhibits a substantial $\Delta E_{Cu-N-C}$ but also offers abundant reaction sites that facilitate catalytic reactions.

## In-situ water remediation using DOP devices

The ability of our decoupled approach to spatially isolate the cathodic and anodic half-reactions provides a feasible way of implementing the high oxidative potentials required for on-site chemical water

treatment of recalcitrant organic compounds. We developed a floating DOP device to validate this capability (Fig. 4a). In the device, a cylindrical glass tube covered with PEM at the bottom was used as a reactor for the activation of PDS. Two interconnected Cu-N-C@GF electrodes were immersed in the PDS solution in the cylindrical tube and the open water body at the bottom of the device, respectively. The device was supported with ethyl vinyl acetate (EVA) foam to make it self-floating on the water surface (Fig. 4b). Notably, this design allows bulk water treatment in large water bodies without the need for specifically designed reaction containers for the target water. For instance, we demonstrated the complete removal of BPA (5 μM) in 2 L of water in 3 h when 40 mL of PDS solution (10 mM) was added to the reactor (Fig. 4c).

Since the reaction only occurs at the co-existence of PDS and BPA (Supplementary Fig. 24), the device possesses a self-responsive characteristic that allows it to respond to changes in the concentration of organic contaminants. To illustrate this, we simulated an application of our method in a consecutive water treatment (Fig. 4d). At the beginning, a device filled with 40 mL of PDS (10 mM) solution was floated on a water tank with 2 L clean water. Before addition of BPA into the tank, the PDS concentration in the reactor was almost unchanged. Once BPA (5.0 μM) was added into the water tank at a reaction time of ~3 h, the concentrations of BPA and PDS spontaneously decreased until the

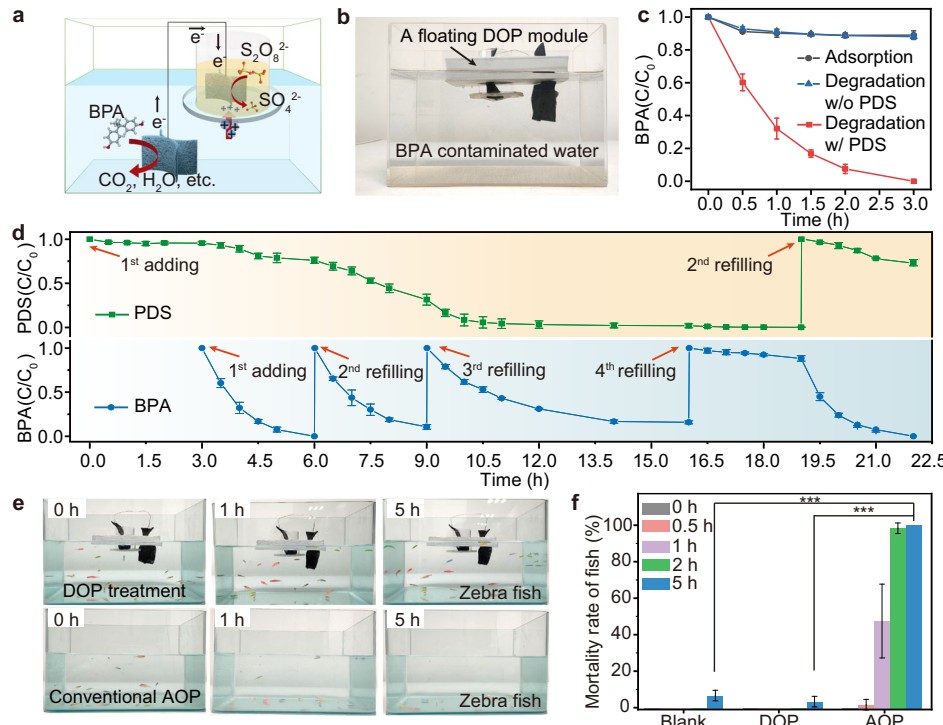

**Fig. 4 | Design of the floating modular device and investigation of its self-responsive water treatment performance. a** Schematic diagram of the floating modular device. **b** Photograph of the floating device setup placed on a water tank. Conditions: PDS, 10 mM, 40 mL; BPA, 5 µM, 2 L. **c** Monitoring of the corresponding changes in BPA concentration within the tank under different conditions. **d** Monitoring of the concentration changes of PDS and BPA during a designed consecutive reaction with scheduled addition of PDS and BPA at specific reaction times. Initial concentration: BPA, 5 µM; PDS, 10 mM; Each refilling: BPA, 5 µM or PDS, 10 mM. **e** Photographs of the water tank captured at scheduled intervals during the BPA degradation experiment, involving treatment with a DOP device and conventional AOP, respectively. **f** Monitoring the mortality rate of zebrafish corresponding to the experiment conducted in **e**. The blank sample was based on zebrafish kept in water containing 5 µM BPA and NaCl 0.5 wt‰ without PDS. The error bars in f were based on parallel experiments of three groups of zebrafish. ***p < 0.001, **p < 0.01, or *p < 0.05. Reaction conditions: BPA, 5 µM; PDS, 1 mM (in the tank) or 10 mM (40 mL, in the tube); NaCl, 0.5 wt‰. Experiments in **c, d** and **f** were conducted in triplicate, and the error bars represent the arithmetic mean ± standard deviation.

complete depletion of BPA. The device was able to respond to the presence of BPA until the stored PDS was completely consumed. As the PDS concentration decreased, the degradation rate of the refilled BPA also decreased because the cathodic reaction in the reactor slowed down (Supplementary Fig. 25). It should be mentioned that after about 19 h of reaction time, when the reactor was filled with a fresh PDS solution, the rate of BPA removal almost returned to its initial state.

We next assessed the effects of our DOP device on aquatic ecosystems by conducting BPA removal in a simulated natural aquatic environment inhabitted by zebrafish. We compared zebrafish mortality rates while treating the water using DOP, traditional heterogeneous AOP, and only adding PDS (Fig. 4e, f and Supplementary Figs. 26 and 27). Notably, introducing 1 mM PDS directly led to rapid and almost total zebrafish mortality within two hours. In contrast, the traditional heterogeneous AOP, in which PDS was added directly in the tank (1 mM, 4 L) with a floating Cu-N-C@GF, exhibited a slower decline in zebrafish vitality due to the catalytic decomposition of PDS. However, all zebrafish died within five hours. In stark contrast, the use of floating DOP devices containing PDS (10 mM, 40 mL) resulted in negligible zebrafish mortality (Fig. 4f, Supplementary Fig. 27)[50]. These results unambiguously indicate that our approach is safe for practical in-situ treatment of contaminated surface water streams. Our floating device exhibited good efficacy in degrading organic pollutants at lower concentrations (20, 2, and 0.2 µg L⁻¹) and in treating a real river water sample (Supplementary Figs. 29 and 30). These results underscore the promising practical application in authentic natural surface water sources, which are characterized by consistently low concentrations of organic pollutants[51,52].

## Viability of the DOP device for large-scale water treatment

More intriguingly, the DOP devices can act as modular units that can be readily integrated for large-scale water treatment (Fig. 5a). In this study, we demonstrated that the integration of six modular devices was able to treat 12 L of contaminated water (BPA conc. 5 µM) at a BPA removal rate of ~0.68 h⁻¹ (Fig. 5b). The BPA removal rate of integrated units in the scale-up treatment is comparable to a single unit working at a smaller scale. However, in practical application, the removal rate may vary depending on the type and concentration of organic pollutants. We also designed an enclosure for the device to facilitate the portability of the devices for outdoor use (Fig. 5c).

To assess the practical utility of our modular device in large-scale water treatment, we conducted a 6-day experiment using a 200 L water tank contaminated with 3 µg L⁻¹ of BPA (as a representative environmentally detected concentration)[53,54]. A single floating modular device was deployed for treatment (Fig. 5d). During the experiment, the BPA concentration in the 200 L water gradually declined in conjunction with the decreasing PDS concentration within the floating module (Fig. 5e). Ultimately, the BPA concentration in the tank dropped to below 0.1 µg L⁻¹, below the Predicted No Effect Concentration (PNEC) established by the U.S. Environmental Protection Agency (EPA) in its BPA Action Plan (2010)[52]. In contrast, a control experiment with the same initial BPA concentration (3 µg L⁻¹) but without the device exhibited no change, affirming that the reduction of BPA in the 200 L water was solely attributed to the implementation of our device. The pH and Cu²⁺ ion concentration of the 200 L water before and after the 6-day experiment were also measured (Fig. 5f, g). The similar pH value and undetected Cu²⁺ further suggest the safety of our device for environmental application.

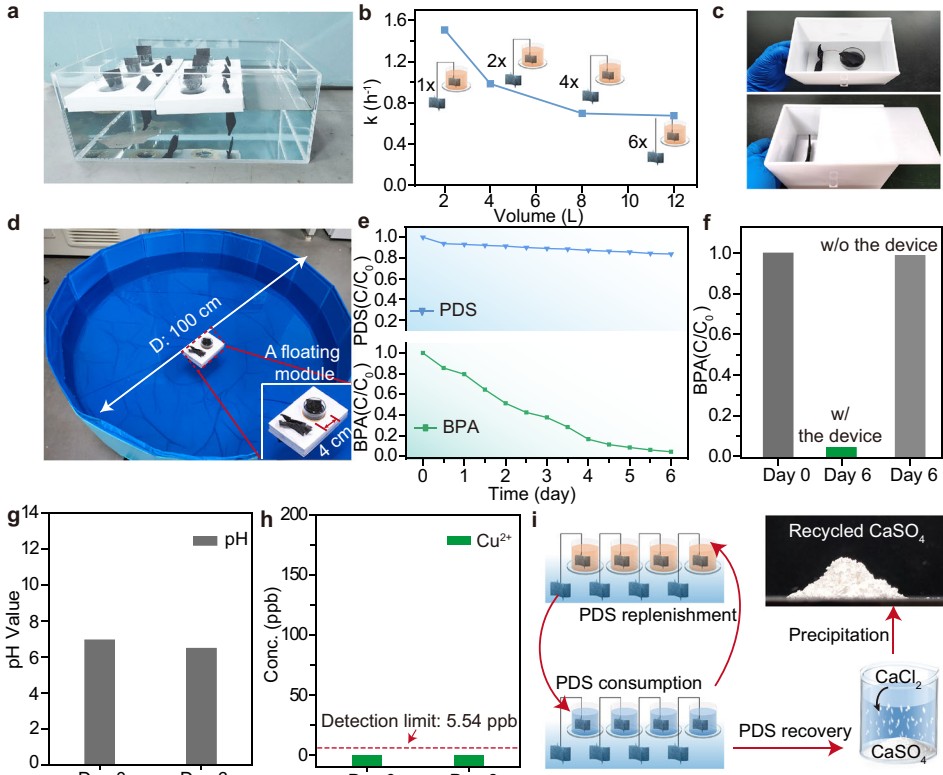

**Fig. 5 | Feasibility of DOP device for in-situ wastewater treatment and resource recovery. a** Photographs of six assembled modules used for treatmenting 12 L wastewater. **b** Calculated degradation rate for BPA wastewater with different volumes and module numbers. Note that the number of the modules were pro-portional amplified in relation to the enlarged volume of wastewater. Reaction conditions: BPA, 5 μM; PDS, 10 mM (40 mL in the tube); NaCl, 0.5 wt‰. **c** Photographs of an enclosure designed for facilitating the portable outdoor application of the device. **d** Photograph of a tank with 200 L BPA wastewater being treated with a floating modular device. **e** Measurement of the concentration changes of PDS in the upper cylindrical tube of the module and BPA in the tank from d, during a continuous 6-day experiment. Conditions: BPA, 3 μg L$^{-1}$; PDS, 10 mM (40 mL in the tube); NaCl, 0.5 wt‰. **f** Monitoring the change in BPA con-centration in the 200 L BPA wastewater before and after the 6-day experiment, with and without the floating device. **g** Detected Cu$^{2+}$ concentration in the 200 L water before and after the 6-day continuous treatment. **h** Measured pH values of the water before and after the 6-day continuous treatment. **i** Schematic diagram showing the cycled consumption and replenishment of PDS during the DOP. The photograph at the top right presented the collected CaSO$_4$ white powder recov-ered from the used PDS waste.

Benefiting from decoupled PDS reduction in isolated reactors, our modular devices enable long-term on-site water treatment through regular PDS replacement. Unlike conventional AOPs where the PDS is directly added to the target water and results in harmful by-products, our approach enables the straightforward recovery of sulfate from the used PDS (Fig. 5i). The collected CaSO$_4$ salt can serve as a versatile raw material in applications such as cement fabrication, agricultural ferti-lizers, and paper manufacturing additives, promoting a sustainable, low-footprint water treatment strategy.

Moreover, treating approximately 200 L of polluted water only requires 20% of the initially added PDS (Fig. 5e). This implies the potential to treat approximately 1000 L (1 m$^3$) of impaired water over an extended period with complete PDS consumption. Based on these findings, it is possible to treat 2500 m$^3$ of contaminated water with 100 floating modular devices, each filled with 1 L of 10 mM PDS. Compared to conventional techniques for similar off-site treatment, our approach is more convenient and costs approximately 4% of ex-situ treatment (Supplementary Fig. 32, see Supplementary Tables 6 and 7 for detailed calculations).

## Discussion

The unique features of the DOP approach render it an ideal candidate for in-situ water treatment in remote or isolated regions where centralized water treatment is not feasible. Notably, this is applicable to rural water sources, encompassing rural springs, local ponds, or reservoirs. More-over, our approach, characterized by its portability and absence of environmental hazard release, exhibits significant potential in addres-sing accidental pollution events. This holds particular relevance in the context of natural water reservoirs, such as small ponds and wetlands, which serve as critical habitats for aquatic organisms. The self-responsive feature of our approach allows for the exclusive utilization of oxidants, conveniently stored on the water surface as a proactive and preventive measure. This introduces a novel, effective, and environmentally-friendly in-situ water treatment option, particularly suited for remote or off-grid areas cohabited by aquatic organisms.

## Methods

### Synthesis of Cu(BTC)(H$_2$O)$_3$ MOF

Cu(BTC)(H$_2$O)$_3$ MOF was synthesized according to the literature[40]. Typically, 2.092 g of Cu(CO$_2$CH$_3$)$_2$•H$_2$O and 0.771 g of L-glutamic acid were added and dissolved well in 500 mL of DI water. At the same time, 1.16 g 1,3,5-benzenetricarboxylic acid was dissolved in a mixed solution with 450 mL DI water and 50 mL ethanol, under ultrasonic sonication. The two solutions were then mixed together and stirred for 2 hours at ambient conditions. After that, the mixture was centrifuged and the obtained blue powder was washed three times with DI water before drying at 60 °C for 12 h. The solid powder obtained was stored for further experiments.

### Synthesis of Cu-N-C

The Cu-N-C was fabricated using Cu(BTC)(H$_2$O)$_3$ MOF as starting material. In a typical experiment, 0.2 g of the Cu(BTC)(H$_2$O)$_3$ MOF

powder was mixed with 2 g of DCD. The mixture was ground in a mortar for 10 min. The sample was calcined at 800 °C for 5 h with a ramping rate of 5 °C min⁻¹ under an argon atmosphere. Finally, the obtained powder was immersed in 40 mL HCl aqueous solution (5 wt%) with saturated oxygen for 4 h to remove residues. After that, the collected powder was washed several times with DI water for, and the final powder was denoted as Cu-N-C and used directly for further experiments and characterization.

## Fabrication of the Cu-N-C@GF electrode

In a typical experiment, the purchased commercial graphite feltswith a thickness of 2.0 mm were successively washed with acetone, ethanol and DI water. After drying at 60 °C for 6 hours, the cleaned GF was shaped into the desired structure for further use. To create the coating, 200 mg of fabricated Cu-N-C was mixed well with 20 mL ethanol, 20 mL DI water, and 200 µL Nafion solution (5 wt%). The mixture was sonicated for 1 h to obtain a stable suspension. Following this, the suspension was evenly dropped onto the prepared GF substrate, which measured 3.0 cm in length, 2.0 cm in height, and 0.20 cm in thickness. The obtained sample was dried at 40 °C for 6 h and subsequently washed several times with DI water. The cleaned sample was then used directly for further experiments.

## Design of the double-chamber reaction

The double-chamber reaction was conducted using a two-chamber reactor with the interconnecting section sealed with the proton exchange membrane. In a typical degradation experiment, 40 mL of 5 µM BPA solution and 40 mL of a 10 mM PDS solution were added respectively into the anodic (right) and cathodic (left) chambers. Two customized Cu-N-C@GF electrodes (3.0 cm × 2.0 cm × 0.20 cm, length × height × thickness) were immersed into the solutions, respectively, with the upper ends connected by a copper wire. At scheduled reaction time intervals, 1 mL of the solution was withdrawn from the anodic chamber and filtered with a 0.22 µM PTEE membrane before being tested by high-performance liquid chromatography (HPLC). Experiments were conducted in triplicate, and the error bars represent the arithmetic mean ± standard deviation.

## Design of the integrated floating modular device

In a typical experiment, one end of a hollow cylindrical glass tube was sealed with PEM, and placed on the surface of a water tank (2 L of 5 µM BPA solution) with the assistance of EVA foam. 40 mL of 10 mM PDS solution was added into the cylindrical tube. Two Cu-N-C@GF electrodes connected with copper wire were immersed into the upper PDS solution and the lower BPA solution, respectively. At the scheduled reaction time intervals, 1 mL of the solution was withdrawn from the anodic chamber and filtered with a 0.22 µM PTEE membrane before being tested by HPLC. Experiments were conducted in triplicate, and the error bars represent the arithmetic mean ± standard deviation.

## Analysis of the harmful effect of oxidants on co-living zebrafish

Adult zebrafish (about 3 months old) were purchased from a local fish shop and 180 live adult zebrafish were randomly assigned into three groups for the parallel experiments. For a completed experiment run, three water tanks were filled with 4 L of water containing 0.5 wt‰ NaCl and 5 µM BPA at room temperature (25 °C) and marked as Tanks I, II and III. A floating tube containing a PDS solution (10 mM, 40 mL) was placed on the water surface of Tank I, while 1.08 g PDS powder was added to Tank II and dissolved well to obtain water with a PDS concentration of 1 mM. Tank III was kept as an blank control group. Subsequently, three groups of twenty adult zebrafish were added to each of the three water tanks. During the 5-hour reaction, the number of dead zebrafish was counted at scheduled time intervals. Experiments were conducted in triplicate, and the error bars represent the arithmetic mean ± standard deviation.

## Concentration measurement of organic compounds

The concentration of organic compounds was analyzed by high-performance liquid chromatographyon a Waters e2968 instrument equipped with a UV detector and a Sunfire C18 4.6 × 250.0 mm reverse-phase column (Waters, USA). The HPLC analysis parameters for different pollutants were listed in Supplementary Table 1 of the Supplementary Information.

## Measurement of PDS concentration

The PDS concentration was analyzed using a colorimetric method[55]. Generally, the presence of NaHCO₃ to maintain the pH of the solution, $S_2O_8^{2-}$ in the solution reacts with KI (see Eq. (4)) to form iodine, which is yellowish in color and has an absorption peak at 352 nm. In a typical experiment, a mixed solution containing 2 g L⁻¹ NaHCO₃ and 41.5 g L⁻¹ KI was freshly prepared. For the detection of PDS, 0.1 mL of collected water sample was added to 4.9 mL of the above-prepared solution, mixed well and allowed to stand for 20 min to complete the reaction. After that, the absorption peak at 352 nm of the yellow solution was recorded using a CARY5000 spectrophotometer. Experiments were conducted in triplicate, and the error bars represent the arithmetic mean ± standard deviation.

$$S_2O_8^{2-} + 2I^- \rightarrow 2SO_4^{2-} + I_2 \qquad (4)$$

## Electrochemical analysis

Electrochemical measurements were conducted via a computer-controlled potentiostat (CHI660E, C17171). A silver/silver chloride electrode (Ag/AgCl), a carbon rod electrode and Cu-N-C@GF were used as reference electrode, counter electrode and working electrode, respectively. The electrolyte was 40 mM boric acid buffer with a pH of 7.4. Cyclic voltammograms were obtained in 0.1 mM pollutant solution at a scan rate of 100 mV s⁻¹. The open circuit potential was monitored by chronopotentiometric analysis upon addition of 1 mM PDS or 0.1 mM pollutants. The corresponding currents were recorded when an electrical potential was set at +0.37 or +0.77 V by measuring chronoamperometry (constant potential) with/without 0.1 mM pollutants in the electrolyte.

## Density functional theory (DFT) calculation

All calculations were carried out using the Vienna Ab initio simulation package (VASP 5.4.1) based on density functional theory[56,57]. The generalized gradient approximation (GGA) with Perdew-Burke-Ernzerhof (PBE) functional was adopted to describe the exchange and correlation potential energy. We chose the projected augmented wave (PAW) potentials to describe the ionic cores and take valence electrons into account using a plane wave basis set with a kinetic energy cutoff of 400 eV. The DFT-D3 empirical correction method was employed to describe van der Waals interactions. Geometry optimizations were performed with a force convergence of smaller than 0.05 eV Å⁻¹. Monkhorst-Pack k-points of 1 × 1 × 1 were applied for all calculations. All atoms were relaxed in the calculations. The adsorption energy ($E_{abs}$) is defined as:

$$E_{abs} = E_{total} - E_{slab} - E_{reference} \qquad (5)$$

## Reporting summary

Further information on research design is available in the Nature Portfolio Reporting Summary linked to this article.

# Data availability

The authors declare that all the data supporting the findings of this study are available within the article (and Supplementary Information Files), or available from the corresponding author on request. All the data generated in this study have been deposited in the

Figshare database under [https://doi.org/10.6084/m9.figshare.24648198][58]. Source data are provided with this paper.

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

## Acknowledgements

We thank the funding support by National Natural Science Foundation of China (22376180, 22176170, 21976152, 22005266), The Key Research and Development Program of Zhejiang Province (2024C03228), Natural Science Foundation of Zhejiang Province (LR21B07002, LR21E020003), "the Fundamental Research Funds for the Central Universities" (2021FZZX001-09) and the Open Research Program of Key Laboratory of 3D Micro/Nano Fabrication and Characterization of Zhejiang Province, Westlake University.

## Author contributions

Z.Y. conducted the experiment, data analysis, and wrote the original manuscript. X.J. conducted the experiment and data analysis. Y.G. contributed to the experiment and manuscript formating. Q.L. contributed to the data analysis. W. X. contributed to the experiment. S.Z., J.Y.W. and D.Y. contributed to the discussion. H.W. contributed to the experiment discussion and data analysis. J.W. conceived and supervised the whole project, wrote and revised the manuscript.

## Competing interests

The authors declare no competing interests.
