## [Peer Review File · Nature Communications]

Decoupled oxidation process enabled by atomically dispersed copper electrodes for in-situ chemical water treatmentReviewers' Comments

Reviewer #1 (Remarks to the Author):

This work essentially consisted of two parts: the electrochemistry part and the "decoupled AOP device" part. The electrochemistry part is highlighted by the development of the Cu-N-C@GF electrode with high sorption sites and low current leakage, as well as the selective ion transport between the solvent side and the oxidant side by a proton exchange membrane (PEM). The first part is an intriguing technique as it combines the strengths of the electrocapacitive process and the electro-oxidation process. The second part involves the construction and testing of a floating reactor for the purification of impaired surface water.

Both parts are worthy of being studied and presented with much more depth. But when combined, the paper becomes unfocused. It is neither scientifically strong (the first part) nor is the applicability of the reactor for surface water purification compelling and practical.

The main criticism of the paper is the practicality of the proposed idea of applying the decoupled AOP system for purifying a body of surface water. The idea of in-situ, floating reactor on the surface of the water body is perceived as a variation of an independent process. It functions in the identical role of treating the impaired water off-site (typically, the treatment station would be isolated in the corner of the water body), followed by circulating the treated water back to the water body. The paper does not justify the competitiveness of such in-situ, floating ideas concerning costs, maintenance, and scalability. Despite the investigators' attempted demonstration of applying the decoupled devices to purify the modeled water in a small tank, the scale is far from reality for the idea to be justified.

Again, the title of the paper deviates from the technical focus of the electrochemical system. The rhetoric of "hazard-free, in-situ" isn't selling very well for the type of application that the authors proposed. In fact, the paper is littered with such rhetoric that the worthy scientific discussion is diluted.

Instead, I suggest that the authors revise their manuscript to focus on the materials development, parameter optimizations, reaction rate, and the fundamental chemistry of the reactions in the experiments. The device itself is so much more intriguing than the application proposed.

Reviewer #2 (Remarks to the Author):

In this manuscript, the authors reported the design of a decoupled advanced oxidation process with the use of a single-atom copper modified graphite felt electrode (Cu-N-C@GF). The process was claimed "hazard-free" for in situ chemical water treatment, in comparison to the conventional advanced oxidation process (AOP). Although the concept sounds intriguing, it may be of limited practical use.

Further, despite the extensive characterization performed by the authors, several key parts are missing from the experimental design. Thus, I would not recommend the acceptance of this manuscript for publication in Nature Communications. Below are some of my general comments.

1. The authors fabricated a delicate Cu-N-C@GF electrode to achieve the proposed decoupled AOP. Would the use of such a delicate and expensive electrode be necessary and economically feasible at all, and do we truly need single-atom Cu? Could the authors just use other less expensive electrodes for proof-of-concept?

2. The authors used bisphenol A (BPA) as a model pollutant to study the decoupled AOP with a BPA concentration of 5 μM . A few other phenols were also studied. Would 5 μM of BPA be relevant to any contaminated water or wastewaters? The authors should focus on more “realistic” pollutants with environmentally relevant concentrations rather than the well-studied model BPA and phenols. For instance, chemical oxidation demand (COD) would be a more suitable parameter for real wastewaters. Further, the authors mainly reported the removal of BPA, but with very limited focus on the end products. For complete pollutant degradation, mineralization rather than removal of the parent compound should be considered. What was the mineralization rate of BPA (how much BPA was fully mineralized under the experimental condition)? How did the toxicity of the end products compare to BPA?

3. The experiments were performed in DI water spiked with BPA, and the complexity of actual water and wastewaters was not considered. Common water constituents, particularly dissolved organic matter can affect the efficiency of the proposed decoupled AOP. In addition, would the interaction between dissolved organic matter and electrode result in the formation of any undesired byproducts that can have adverse effect on the ecosystem?

4. The zebra fish toxicity study was not well designed. It is common sense that strong oxidants like peroxysulphate (PDS) would have detrimental effect on aquatic organisms and it is fully expected that direct addition of PDS would cause zebra fish death. The authors showed that the decoupled AOP did not cause the death of zebra fish, but that does not necessarily mean the decoupled AOP is safe for practical application. First, as specified in a previous comment, the authors should compare the toxicity of the end products versus the parent compound, not with the unrealistic conventional AOP process. Second, the long-term toxicity should also be considered in addition to the acute toxicity.

Reviewer #3 (Remarks to the Author):

The authors reported a decoupled AOP system consisting of a floating device for the degradation of organic pollutants in a hazard-free manner. They evaluated the performance of a designed device by testing the degradation of various contaminants and identified the active site of the catalyst (Cu-N-C@GF) via computational analysis. They also conducted an environmental impact assessment of PDS

(oxidant). Finally, they investigated the feasibility of the integrated decoupled AOP device for potential in-situ wastewater treatment and resource recovery. This research is very interesting and opens up a lot of possibilities for water treatment. However, this manuscript is difficult to accept in its current form and requires major revision. The detailed comments are as follows.

1. Why did the authors use the Cu-N-C@GF catalyst? This needs to be explained in the introduction.
2. Can the Cu-N-C@GF catalyst itself activate PDS? It is necessary to evaluate the PDS activation efficiency of the catalyst itself without using a decoupled AOP system. If PDS is activated on the catalyst, it is doubtful that a decoupled AOP system is necessary.
3. Did the author test BPA degradation in a reactor without PEM? I am not sure if PEM is really necessary in this system.
4. Why was NaCl (0.5 wt%) added into the reactor? What is the role of NaCl? Is it possible reactive chlorine species to participate in the BPA degradation reaction? I don't understand why NaCl was added.
5. This is a similar question to comment 2. In Figure 3e, if PDS is added to a tank with a device, can PDS be catalytically activated? If this happens, the authors should compare the mortality rate of zebra fish when PDS is added in a tank with a modular device. While the authors have confirmed that PDS has an adverse effect to zebra fish, it remains to be seen how PDS affects zebrafish when it is activated by the catalyst.
6. When the catalyst was used for a long time or repeatedly, did the authors check for possible leaching of Cu ions from the catalyst? If leaching occurs, the authors need to confirm that leaching affects zebra fish.
7. Did the authors have a specific reason for using low concentration of BPA (5 μM) and high concentration of PDS (10 mM)? Did the authors use such extreme concentrations of BPA and PDS because of the low efficiency of the catalyst?

Responses to the reviewers' comments (Manuscript number NCOMMS-23-14745)

Changes in the revised manuscript as a response to the reviewers' comments are highlighted in red color and clarifications regarding the reviewer's comments are provided in blue color.

Reviewer #1 (Remarks to the Author):

This work essentially consisted of two parts: the electrochemistry part and the "decoupled AOP device" part. The electrochemistry part is highlighted by the development of the Cu-N-C@GF electrode with high sorption sites and low current leakage, as well as the selective ion transport between the solvent side and the oxidant side by a proton exchange membrane (PEM). The first part is an intriguing technique as it combines the strengths of the electrocapacitive process and the electro-oxidation process. The second part involves the construction and testing of a floating reactor for the purification of impaired surface water.

Both parts are worthy of being studied and presented with much more depth. But when combined, the paper becomes unfocused. It is neither scientifically strong (the first part) nor is the applicability of the reactor for surface water purification compelling and practical.

Response: We would like to express our gratitude to the reviewer for acknowledging the value of our work and providing insightful suggestions for its enhancement.

It is true that our manuscript mainly contains two sections: the electrochemistry part and the "decoupled AOP device" part, the electrochemistry part demonstrates the fundamental requirement for a "decoupled AOP" but both parts are essential for the demonstration of our at the electrochemistry part, we introduced the design concept of our "decoupled AOP" process, which is based on the design the two parts connected closely and both contribute to the new concept we would like to deliver in this manuscript. We agree with the reviewer that the proposed process should be studied with more depth. Herein, in the revision we have tried our best to provide further experimental demonstrations and more in-depth discussion to strength the claim of our work. And all the details can be found in the following point-by-point responses, as well as in the revised manuscript. For easier understanding, we listed below the main changes we conducted in the revised manuscript below.

The main updates and supplements in the revised version to elaborate the science in the technique:

1. Scheme 1 of the revised manuscript has been revised, to illustrate better the reaction mechanism difference between decoupled AOP and conventional AOP.

2. An in-depth discussion on the reaction mechanism has been added, starting from the powder Cu-N-C based conventional AOP reaction to the derived decoupled system, as well as the optimization of the synthesis parameters for Cu-N-C (Supplementary Figure 12-14).

3. Adoption of other commercial catalysts (e.g. CuO, CNTs) to demonstrate the generality of the decoupled AOP strategy (Figure 2g and h, Supplementary Figure 22 and 23).

The main supplements to demonstrate the potential applicability of our reactor:

1. A systematical investigation about the impact of common inorganic ions existed in natural surface water, natural organic matter (NOM), and pH fluctuations on the system performance have been added (Supplementary Figure S20).

2. Performance stability of the system has been evaluated by conducting a ten-times cycled oxidization of BPA (Supplementary Figure S 21).

3. The COD value change of the oxidized products and the corresponding toxicity analysis of the detected compounds have been conducted (Supplementary Figure S8 and 9).

4. A comparative analysis of zebrafish mortality rates during water treatment using decoupled AOP, traditional AOP, and the sole presence of PDS has been conducted (Supplementary Fig. 27).

5. The practical utility of the modular device for potential large-scale water treatment has been investigated via conducting a 6-day experiment using a 200 L water tank contaminated with BPA (Figure 4d-g).

The main criticism of the paper is the practicality of the proposed idea of applying the decoupled AOP system for purifying a body of surface water. The idea of in-situ, floating reactor on the surface of the water body is perceived as a variation of an independent process. It functions in the identical role of treating the impaired water off-site (typically, the treatment station would be isolated in the corner of the water body), followed by circulating the treated water back to the water body. The paper does not justify the competitiveness of such in-situ, floating ideas concerning costs, maintenance, and scalability. Despite the investigators' attempted demonstration of applying the decoupled devices to purify the modeled water in a small tank, the scale is far from reality for the idea to be justified.

Response: We thank the reviewer for providing critical comment about the practicality of our proposed AOP. We are sorry that we did not clearly demonstrate the advantage of the process in the original manuscript. We believe the decoupled AOP process (now rephrased as decoupled oxidation process (DOP) in revised manuscript for easier understanding) does contain some advantages compared to conventional off-site water treatments in terms of costs, maintenance, and scalability. Firstly, the conventional off-site approaches always need water pumps to circulate water for treatments, resulting in considerable energy costs (e.g. pumping 2500 m³ of water may necessitate 7800 kWh of electricity, as shown in Table R1 below). These methods are difficult to be applied at remote areas where power supplies and infrastructures are barely implemented, such as wild wetlands, isolated islands, and rural regions. By contrast, our in-situ floating approach can be directly deployed in bulk waterbodies, requiring no external water pumps and electric consumption. It makes this approach potentially promising for low-cost decentralized water treatments in remote and pollution-scattered areas. It offers a viable solution for pollution control in area-source pollution scenarios, where the large volume of polluted water cannot be pumped out due to logistical constraints.

Figure R1. Schematic diagram showing the applying of water treatment station for the impaired isolated waterbodies and our DOP floating device for in-situ water remediation.

Table R1. Comparison of estimated cost for treating 2500 m³ amount of impaired water with different techniques

	Construction cost for 2500 m ³ water treatment ability	Operation cost		Characteristics of operation	Total Cost/2500 m ³ impaired water
		Electricity consumption/m ³ impaired water	Chemical usage/2500m ³ impaired water		
Our floating device with DOP process	Electrodes + Floating set ~\$ 5.3 per set \$ 5.3 × 100 sets = \$ 530 for treating 2500 m ³ impaired water at site ^a	No power consumption	~2.7 kg of PDS, ~\$ 2.7 ^d	in-situ and “self-responsive”	~\$ 500
Conventional AOP water treatment station	Water tank+ Pump + Pipeline + Electric generator + Treatment system >\$ 10,000^b	~3.12 kWhm ⁻³ × ~\$ 0.36 kWh ⁻¹ × 2500 m ³ = \$ 2808^c	~2.7 kg of PDS, ~\$ 2.7 ^e	off-site, need monitoring on the occurrence of pollutants	>\$ 13,000

Note: a. ¹The set used here is the enlarged set which can contain 1 L 10 mM PDS solution. ²The price was calculated based on the materials involved in the device fabrication, including the graphite felts, chemicals for catalyst synthesis, cylinder tube, copper wire and etc.

b. Prices of the equipment were determined based on the average prices on website.

c. The average kWhm⁻³ was calculated based on the average amount given in references (*Sci. Total Environ.* **2021**, 782, 146599).

d. The consumed PDS amount and price was calculated based on the experiment on Figure R1, Table R1 and price searched on website.

e. We estimated the PDS amount in conventional AOP process as the same amount in our DOP process, however, in dealing with trace micropollutants, the conventional AOPs always require oxidants with an overdose amount, due to the radical scavenging effects (*Environ. Sci. Technol.* **2021**, 55, 14494-14514; *Nature* **2016**, 529, 190-194).

In the revision, we developed a large-scale water-treatment experiment to further

demonstrate the potential use of our device towards practical large-scale water treatment. We prepared a tank (diameter: 1 m) containing 200 L water and use one floating modular device to treat the BPA (3 $\mu\text{g/L}$) contaminated water (Figure R2 below). The BPA concentration was set as 3 $\mu\text{g/L}$, as a representative value detected in environment (*Toxicol. Environ. Chem.* **2013**, 95, 386-399; *Environ. Sci. Pollut. Res.* **2021**, 28, 23, 30242-30254). The PDS solution volume stored in the floating device was set as 40 mL, in a concentration of 10 mM, same as the device we used in the manuscript. As can be seen, during the 6-day continuous experiment, the detected BPA concentration in the 200 L water decreased gradually, along with the decreased PDS concentration in the device set (Figure 2b). The final BPA concentration in the tank was reduced to less than 0.1 $\mu\text{g/L}$, which was lower than the predicted no effect concentration (PNEC) in the file of U.S. Environmental Protection Agency (EPA) (*U.S. Environmental Protection Agency*, BPA action plan, **2010**). While in contrast, in the control experiment with same initial BPA concentration (3 $\mu\text{g/L}$) but without use of the device, the BPA concentration remains same as the original value, which proves that the removal of BPA in the 200 L water can be fully attributed to the employment of our device (Figure R2c).

Additionally, as unveiled in Figure 2d, treatment of the ~ 200 L impaired water only consumes 20% of the added PDS. Thus, we can propose that it is possible to treat $\sim 1,000$ L (1 m^3) impaired water in a longer period with complete consumption of the filled PDS. Importantly, the water treatment capacity by our approach can be further extended to even larger scale by simply applying multiple floating devices on a water surface. Based on the additional findings, it is possible to treat 2500 m^3 of scale water (\sim the size of a swimming pool) with 100 floating water treatment modules with 1 L PDS filled in each. Compared with conventional off-site techniques for the same treatment, our approach is much more convenient and cost effective (as summarized in Table R1 and R2).

Table R2. Parameters of the modular device and water tank involved in the 6-day continuous treatment.

	Diameter (m)	Occupied Area (m^2)	Volume (L)
Modular device	0.04	0.005	0.04
Tank with impaired water	1.5	7.065	200
Impaired water /device	37.5	1413	5000

Besides, pH value and Cu^{2+} ion concentration of the 200 L water before and after the 6-day experiment was also measured (Figure R2d and e). The similar pH value as well as the non-detected Cu^{2+} concentration (below the detection limit) further suggest the safety of our device to the environment application.

Figure R2. The water treatment performance of one modular device towards a relatively large-scale water (200 L). (a) Photograph of the water tank with floating modular device during the treatment. (b) The corresponding measured PDS concentration in the upper cylindrical tube and BPA concentration in the water tank. (c) The BPA concentration change in the water tank with and without the floating device. (d) The detected copper ion concentration and (e) the measured pH value of the water before and after the 6-day continuous treatment.

Figure R2, Table R1 and R2 have been added into the revised manuscript as Figure 4d-h, Table S5 and S6 with a proper description.

1) Again, the title of the paper deviates from the technical focus of the electrochemical system. The rhetoric of "hazard-free, in-situ" isn't selling very well for the type of application that the authors proposed. In fact, the paper is littered with such rhetoric that the worthy scientific discussion is diluted.

Response: We thank the reviewer for the thoughtful comments. We have conducted more experiment and scientific discussion to improve our manuscript. We also rephrased the title to "Decoupled oxidation process enabled by atomically dispersed copper electrodes for in-situ chemical water treatment" to elaborate more clearly the key concept we would like to deliver in the manuscript.

2) Instead, I suggest that the authors revise their manuscript to focus on the materials development, parameter optimizations, reaction rate, and the fundamental chemistry of the reactions in the experiments. The device itself is so much more intriguing than the application proposed.

Response: We thank the reviewer for the critical suggestion. We have performed sets of

additional experiments to elaborate more on the material development, parameter optimizations and fundamental mechanism analysis. Please refer to the following sections for the detailed information (Figure R3-R9 below). The manuscript and the Supplementary Information have also been revised with a proper addition of the new data.

As we elaborated in the manuscript, the generated positive potential difference (ΔE) between the cathodic (PDS+Cu-N-C@GF electrode) and anodic chamber (BPA+Cu-N-C@GF electrode) mainly served to drive the transferring of electrons from PDS to BPA to subsequently trigger the degradation of BPA. Since in our previous investigation that the unmodified graphene felt (GF) electrodes exhibit negligible efficacy in the removal of BPA (Figure S9 of the original manuscript which is now Figure S11 of the revised version). We thus attribute the remarkable catalytic removal of BPA to the Cu-N-C.

We conducted additional experiments to further elucidate the role of Cu-N-C electrodes. The physical absorption of Cu-N-C to BPA was measured at the first. As shown by Figure R3a below, pure physical absorption of BPA didn't significantly contribute to its removal, thus we conclude that the catalytic activation of PDS dominates the BPA removal. As the catalytic activation of PDS always generate oxidative radicals, we added different scavengers during the catalytic removal process (Figure R3b). The addition of dimethyl sulfoxide (DMSO) and methanol (MeOH) showed little influence on BPA removal, suggesting that the oxidation of BPA in the PDS/Cu-NC system did not rely on the radical species $\text{SO}_4^{\cdot-}$ and $\cdot\text{OH}$ (*Appl. Catal. B: Environ.* **2017**, 210, 444-453). Benzoquinone (BQ) and L-histidine, as typical trapping agents for $\text{O}_2^{\cdot-}$ and $^1\text{O}_2$, didn't effectively inhibit bisphenol A removal, suggesting that radical species $\text{O}_2^{\cdot-}$ and $^1\text{O}_2$ were also not responsible for the reaction (*Angew. Chem. Int. Ed.* **2021**, 60, 4588-4593). A further EPR analysis results presented on Figure R3c also unveiled the absence of $\text{SO}_4^{\cdot-}$, $\cdot\text{OH}$, $\text{O}_2^{\cdot-}$ and $^1\text{O}_2$ in the reaction, matching well with the conclusion in Figure R3b.

Further electrochemical analysis was also conducted in the revision. We studied the open circuit potentials of the electrode in different conditions by chronopotentiometry (Figure R3d). As can be seen, there is no obvious potential difference between the sole PDS and BPA, however, the presence of Cu-N-C helped to create an obvious potential difference ($\Delta E=0.9-0.34=0.56$ V) between the two ends (Cu-N-C+PDS) and (Cu-N-C+BPA). Taking the above phenomenon into consideration, we can propose that once BPA and PDS met Cu-N-C, they will be absorbed on Cu-N-C and formed a positive ΔE between the absorbed PDS* and BPA*. The positive ΔE will can drive the electrons to transfer from the BPA* to PDS* ends, resulting in the oxidation of BPA (shown as the schematic diagram in Figure R3e). Benefiting from the electron transfer from PDS to BPA with Cu-N-C as the bridge, a physical separation of PDS but connect them electrically can still persist the positive potential difference, and thus supporting the DOP reaction. A schematic diagram for the transformation from powder catalyst based redox reaction to DOP was shown in Figure R4 below.

Figure R3. Performance measurement and analysis with the direct mixing of Cu-N-C composite, PDS and BPA in a one-pot reaction. (a) The monitored BPA concentration change under different conditions. Conditions (if required): catalyst = 0.15 g/L, BPA = 0.1 mM, PDS = 1 mM. (b) Monitored BPA removal performance under the presence of difference scavengers. Conditions (if required): catalyst = 0.15 g/L, BPA = 0.1 mM, PDS = 1 mM, L-histidine = 50 mM, Methanol = 500 mM, DMSO = BQ = 5 mM. (c) The measured EPR spectra with DMPO and TEMP as the trapping agent, respectively. Conditions (if required): catalyst = 0.15 g/L, BPA = 0.1 mM, PDS = 1 mM, DMPO = TEMP = 50 mM. (d) The Open-circuit potential curve measurement of BPA (0.1 mM) and PDS (1 mM) solutions with different triple electrode system. Reference electrode: silver/silver chloride electrode (Ag/AgCl). Counter electrode: platinum electrode. Working electrodes: glassy carbon electrode, and glassy carbon electrode coated with Cu-N-C. (e) Scheme illustration of the electron transfer process involved in the oxidation of BPA under the assistance of powder Cu-N-C catalyst and PDS.

Figure R4. Schematic diagram of design ideas from the catalyst powder to the device.

The above analysis confirmed the role of Cu-N-C for the catalytic reaction. Since it is widely reported that the calcination temperature always influences the cheating environment of atomically dispersed metals and thereafter the functionality (e.g. *Angew. Chem. Int. Ed.* **2018**, *57*, 1944; *ACS Energy Lett.* **2020**, *5*, 1044; *Angew. Chem. Int. Ed.* **2018**, *57*, 1944). The calcination temperature of the prepared precursor (Cu(BTC)(H₂O)₃ MOF and DCD) was set as 600 °C, 700 °C and 900 °C, respectively. The obtained composites were named as Cu-N-C-6, Cu-N-C-7 and Cu-N-C-9, accordingly. Since the catalyst Cu-N-C we used to demonstrate our

design concept was calcined at 800 °C, it was remarked as Cu-N-C-8 here for easier understanding. The scanning electron microscope (SEM) images of the four obtained composites were presented in Figure R5a-d. As can be seen, elevating the calcination temperature from 600 to 800 °C enhances the material's porosity, while a subsequent increase to 900 °C causes the porous structure to collapse. The subsequent N₂ adsorption-desorption isotherms measurement of the Cu-N-C-n (n=6, 7, 8, 9) series unveiled that Cu-N-C-8 owned the largest specific surface around 200 m² g⁻¹, matching well with the above SEM analysis. X-ray diffraction (XRD) patterns of the samples only present the peaks related to the graphitic carbon structure, which suggests that Cu dispersed atomically on the matrix. The additional Raman analysis on Figure R5g showed an increased I_D/I_G value from Cu-N-C-6 to 9, suggesting a higher graphitization degree along with the pyrolysis temperature. Via the thermogravimetric analysis (TGA), it clearly showed that Cu-N-C-8 owned the highest Cu loading density, which is about approximately 16.37 wt%.

The four composites were adopted to catalyze the oxidation of BPA with mixing with PDS and BPA respectively in an all-in-one solution-based reaction system (Figure R5i and j). As can be seen, Cu-N-C-8 showed an excellent BPA removal performance close to 100% within 4 mins, with a *k* value around 1.78 min⁻¹. Its performance is not only the best among the four composites, but also much higher than most of the reported literatures (see Table R3 below). Via the above analysis, we can attribute the excellent catalytic performance of Cu-N-C-8 to its porous structure with relatively large surface area, as well as the highest atomically dispersed Cu atoms. Therefore, we can conclude that the atomically dispersed Cu sites may mainly serve as the active reaction sites. Since composite Cu-N-C-8 showed the best catalytic performance, it was selected as the catalyst for further experiment and we simplified its name to Cu-N-C in the following sections.

Figure R5. (a-d) The respective low-resolution SEM images of prepared Cu-N-C-n series (n=6, 7, 8 and 9). (e) The measured N₂ adsorption-desorption isotherms and pore size distributions (the inset graph) of Cu-N-C-n (n=6, 7, 8 and 9). (f) The measured XRD spectra of Cu-N-C-n (n=6, 7, 8 and 9). (g) The

measured Raman spectra of Cu-N-C-n (n=6, 7, 8 and 9). (h) The measured TGA results of Cu-N-C-n (n=6, 7, 8 and 9). The Cu element of catalysts can be calculated by multiplying the rest of mass ratio and the Cu amount of CuO. (i) The monitored BPA concentration change in one-pot reaction with addition of different catalysts. (j) The calculated k values in correspondence of i. Conditions (if required): catalyst = 0.15 g/L, BPA = 0.1 mM, PDS = 1 mM.

Table R3. Comparison of the catalytic performance of Cu-N-C-8 with the reported literatures

Catalyst	Oxidant dosage	Pollutant (mg L ⁻¹)	Removal efficiency	k value (min ⁻¹)	TOF (L g ⁻¹ min ⁻¹)	Reference
Cu-N-C-8 (0.15 g/L)	PDS (1 mM)	BPA (0.1 mM)	100% (5 min)	1.777	11.85	This work
N-BC900 (0.2 g/L)	PDS (2 mM)	BPA (88 μM)	100% (20 min)	0.214	1.07	Environ. Sci. Technol. 2018, 52, 15, 8649
CNF3 (0.10 g/L)	PMS (1 mM)	4-CP (0.1 mM)	100% (20 min)	0.254	2.54	Environ. Sci. Technol. 2018, 52, 4, 2197
Fe-SACs (0.2 g/L)	PMS (1.3 mM)	BPA (0.11 mM)	88% (30 min)	0.104	0.52	Angew. Chem. Int. Ed. 2021, 60, 22513
Fe _{0.15} Mn _{0.85} O ₂ (0.04 g/L)	PMS (0.5 mM)	BPA (5 μM)	100% (21 min)	~0.23	~5.75	Environ. Sci. Technol. 2019, 53, 21, 12610
CuO (0.2 g/L)	PDS (40 μM)	2,4-DCP (5 μM)	100% (40 min)	0.062	0.31	Environ. Sci. Technol. 2014, 48, 10, 5868
NBC-4 (1.0 g/L)	PDS (1 mM)	SDZ (10 μM)	96.2% (60 min)	0.0798	0.0798	Water Res. 2019, 160, 405
Cu-N ₄ /C-B (0.1 g/L)	PMS (0.65 mM)	BPA (88 μM)	98% (5 min)	0.56	5.6	Proc. Natl. Acad. Sci. USA 2022, 119, e2119492119
PSS-800 (0.2 g/L)	PDS (1.85 mM)	2,4-DCP (0.62 mM)	100% (60 min)	0.0606	0.303	Appl. Catal. B: Environ. 2020, 260, 118160
Cu _{SA} -NC (0.04 g/L)	PDS (0.5 mM)	2,4-DCP (0.1 mM)	100% (30 min)	0.284	7.1	Environ. Sci. Technol. 2022, 56, 12, 8765
ZnFeMnO ₄ (0.1 g/L)	PMS (0.16 mM)	BPA (40 μM)	100% (15 min)	0.43	4.3	Proc. Natl. Acad. Sci. USA 2020, 117, 30966
Annealed CNT (0.1 g/L)	PDS (1 mM)	Phenol (0.1 mM)	100% (30 min)	0.198	1.98	Environ. Sci. Technol. 2020, 54, 1267

Note: The turnover frequency (TOF) was calculated through dividing the reaction rate of pollutant degradation by the catalyst concentration.

We tuned the loading amount of Cu-N-C on graphite felt electrode to investigate their influence on the DOP reaction (Figure R6a). As can be seen, a first increase of the loading catalyst amount from 0.3 to 0.6 mg/cm² resulted in significant boosted BPA removal efficiency, however, a further increased loading amount to 1 mg/cm² didn't contribute to obvious improvement. Therefore, the catalyst loading amount 0.6 mg/cm² on the electrode was adopted in our manuscript for the further experiment. We further test the removal efficiency of our reactor towards BPA solution in different concentration (Figure R6b). Our DOP presented a rapid removal of BPA, towards a wide concentration range from 2 μM to 10 μM, which might suggest the application of this set for the treatment towards different concentration of pollutants.

Figure R6. (a) The measured BPA removal performance in the DOP two-chamber reactor, with different catalyst loading amount on the graphite felt electrode. Conditions: BPA = 5 μ M, PDS = 10 mM, NaCl = 0.5 wt%. (b) The measured BPA removal performance in the DOP two-chamber reactor, towards different BPA concentration. Conditions: PDS = 10 mM, NaCl = 0.5 wt%.

We hope the above analysis can help to justify the catalyst design and fundamental chemistry of the reactions more clearly. Besides, to demonstrate the generality of our DOP strategy, we further tried several other catalysts and found that commercial CuO and carbon nanotubes (CNTs) also can help to produce a positive potential difference (although smaller than our Cu-N-C system) on the cathodic and anodic ends (see Figure R7-R9 below). The CuO and CNTs were coated onto the graphite felt (GF) to prepare the CuO@GF and CNTs@GF electrodes, respectively (Figure R7 and R8 below). The low-resolution SEM and XRD analysis were conducted respectively to show the successful coating of CuO and CNTs onto the electrode (Figure R7a-c and Figure R8a-c). By conducting the DOP reaction with the prepared CuO@GF and CNTs@GF electrodes, respectively (Figure R7d and R8d), they both showed BPA removal capability in a certain degree (Figure R7e and R8e). The respective chronopotentiometry analysis unveiled a ΔE_{CuO} around 0.34 V and ΔE_{CNT} around 0.23 V existed in the two reaction systems (Figure R7f and R8f). Furthermore, under the respectively applied voltages, the detected current further support the occurrence of the electrochemical catalytic reaction (Figure R7g and R8g). The above experiment helped to demonstrate the generality of our strategy.

It should be mentioned that, although composite CuO and CNTs can trigger the DOP reaction, our Cu-N-C presented the best organic pollutant removal performance (Figure R9). We attributed this phenomenon to two main reasons, first is the larger $\Delta E_{\text{Cu-N-C}}$ (0.5 V) than ΔE_{CuO} (0.34 V) and ΔE_{CNTs} (0.23 V). Second reason is the abundant atomically dispersed Cu sites on the electrode surface to catalyze the reaction. As we mentioned in the manuscript, the atomically dispersed Cu-N₂ served as the active sites for the catalytic reaction, and the Cu loading density of Cu-N-C is approximately 16.37 wt%, thus the Cu-N-C@GF electrode can provide more active sites for better reaction performance.

Figure R3-9 and Table R3 have been added into the revised manuscript as Figure S12-14, S17, S22, S23, Figure 2g, h and Table S4, with a proper description.

Figure R7. (a) SEM image of CuO@GF. (b) SEM image of CuO. (c) The measured XRD pattern of CuO. (d) Photograph of the double-chamber reactor we set to test performance of CuO@GF. (e) The corresponding measured BPA removal performance with the set in d. Conditions: BPA = 5 μM , PDS = 10 mM, NaCl = 0.5 wt%. (f) The measured open-circuit potential curve on the CNTs@GF. Conditions: BPA = 0.1 mM, PDS = 1 mM. Reference electrode: silver/silver chloride electrode (Ag/AgCl). Counter electrode: a carbon rod electrode. Working electrodes: CuO@GF electrode. (g) Measurement of current-time curves with different potentials on the CuO@GF electrode. Conditions: BPA = 0.1 mM, PDS = 1 mM. Reference electrode: silver/silver chloride electrode (Ag/AgCl). Counter electrode: a carbon rod electrode. Working electrodes: CuO@GF electrode.

Figure R8. (a) SEM image of CNTs@GF. (b) SEM image of CNTs. (c) TEM image of CNTs. (d) Photograph of the double-chamber reactor we set to test performance of CNTs@GF. (e) The corresponding measured BPA removal performance with the set in d. Conditions: BPA = 5 μM , PDS = 10 mM, NaCl = 0.5 wt%. (f) The measured open-circuit potential curve on the CNTs@GF. Conditions: BPA = 0.1 mM, PDS = 1 mM. Reference electrode: silver/silver chloride electrode (Ag/AgCl). Counter electrode: a carbon rod electrode. Working electrodes: CNTs@GF electrode. (g) Measurement of current-time curves with different potentials on the CNTs@GF electrode. Conditions: BPA = 0.1 mM, PDS = 1

mM. Reference electrode: silver/silver chloride electrode (Ag/AgCl). Counter electrode: a carbon rod electrode. Working electrodes: CNTs@GF electrode.

Figure R9. (a) The removal ratio of BPA in the DOP towards different catalysts. Conditions: BPA = 5 μ M, PDS = 10 mM, NaCl = 0.5 wt%. (b) Comparison of potential differences at different electrodes.

Reviewer #2 (Remarks to the Author):

In this manuscript, the authors reported the design of a decoupled advanced oxidation process with the use of a single-atom copper modified graphite felt electrode (Cu-N-C@GF). The process was claimed “hazard-free” for in situ chemical water treatment, in comparison to the conventional advanced oxidation process (AOP). Although the concept sounds intriguing, it may be of limited practical use. Further, despite the extensive characterization performed by the authors, several key parts are missing from the experimental design. Thus, I would not recommend the acceptance of this manuscript for publication in Nature Communications. Below are some of my general comments.

Response: We thank the reviewer for the critical comments. We have performed more experiments to demonstrate the potential implementation of our strategy in wastewater treatment. Please refer to the point-by-point response for the detailed comments below as well as the revised manuscript. We hope that the additional experiments and the response can be useful for further justify the importance of our study.

1) The authors fabricated a delicate Cu-N-C@GF electrode to achieve the proposed decoupled AOP. Would the use of such a delicate and expensive electrode be necessary and economically feasible at all, and do we truly need single-atom Cu? Could the authors just use other less expensive electrodes for proof-of-concept?

Response: We thank the reviewer for the critical comments. Accordingly, we have carefully evaluated the cost of our single-atom electrode and provided further justification about the necessity of the electrode being used.

Firstly, the raw materials involved for the fabrication of Cu-N-C@GF electrode were listed in Table R4 below with their approximate cost. The cost for one Cu-N-C@GF electrode is ~ \$ 0.02, and the recycled decoupled reaction experiment in Figure R10 below unveiled that after a 10-cycled BPA removal experiment, the electrode still can achieve a removal rate close to 90%, thus we think that the price of the Cu-N-C@GF electrode is acceptable for the potential use.

Table R4. The respective calculation of the fabrication cost for different electrode

Material	Unit Price (\$)	Amount/size used for one electrode	Partial cost for one electrode (\$)
Copper (II) acetate monohydrate	4.75/kg	5.08 mg	2.41×10^{-5}
L-glutamic acid	1.40/kg	1.87 mg	2.62×10^{-6}
1,3,5-benzenetricarboxylic acid	4.89/kg	2.81 mg	1.37×10^{-5}
Dicyandiamide	2.79/kg	41.15 mg	1.15×10^{-4}
Cu-N-C	43.13/kg	3.60 mg	1.55×10^{-4}
Nafion solution	1.61/ mL	3.60 μ L	5.80×10^{-3}
Graphite felts	21.00/m ²	6.00 cm ²	1.26×10^{-2}
Estimated cost for one Cu-N-C@GF electrode			~ \$ 0.0186
CNTs	139.70/kg	3.60 mg	5.03×10^{-4}
Nafion solution	1.61/ mL	3.60 μ L	5.80×10^{-3}

Graphite felts	21.00/m ²	6.00 cm ²	1.26×10 ⁻²
Estimated cost for one CNTs@GF electrode			~ \$ 0.0189
CuO	11.32/kg	3.60 mg	4.08×10⁻⁵
Nafion solution	1.61/ mL	3.60 μL	5.80×10 ⁻³
Graphite felts	21.00/m ²	6.00 cm ²	1.26×10 ⁻²
Estimated cost for one CuO@GF electrode			~ \$ 0.0184

Figure R10. Cyclic degradation performance of our Cu-N-C@GF electrode for the BPA removal with the DOP process. Conditions: BPA = 5 μM, PDS = 10 mM, NaCl = 0.5 wt%.

Secondly, as we demonstrated in the manuscript, the main driving force for the decoupled AOP process (now rephrased as decoupled oxidation process, DOP, in the revised manuscript for easier understanding) is the generated positive potential difference (ΔE) between the cathodic and anodic ends. To demonstrate the generality of our strategy, we tried several other catalysts and found that commercial CuO and carbon nanotubes (CNTs) also can help to produce such a potential difference on the two ends (see Figure R11-R13 below). The CuO and CNTs were coated onto the graphite felt (GF) to prepare the CuO@GF and CNTs@GF electrodes, respectively (Figure R11 and R12 below). The low-resolution SEM and XRD analysis were conducted respectively to show the successful coating of CuO and CNTs onto the electrode (Figure R11a-c and Figure R12a-c). By conducting the DOP reaction with the prepared CuO@GF and CNTs@GF electrodes, respectively (Figure R11d and R12d), they both showed BPA removal capability in a certain degree (Figure R11e and R12e). The respective chronopotentiometry analysis unveiled a ΔE_{CuO} around 0.34 V and ΔE_{CNTs} around 0.23 V existed in the two reaction systems (Figure R11f and R12f). Furthermore, under the respectively applied voltages, the detected current further support the occurrence of the electrochemical catalytic reaction (Figure R11g and R12g). The above experiment helped to demonstrate the versatility and generality of our strategy.

It should be mentioned that, although composite CuO and CNTs can trigger the DOP reaction, our Cu-N-C presents the best organic pollutant removal performance (Figure R13). We attributed this phenomenon to two main reasons, first is the larger $\Delta E_{\text{Cu-N-C}}$ (0.5 V) than ΔE_{CuO} (0.34 V) and ΔE_{CNTs} (0.23 V). Second reason is the abundant atomically dispersed Cu sites on the electrode surface to catalyze the reaction. As we mentioned in the manuscript, the atomically dispersed Cu-N₂ served as the active sites for the catalytic reaction, and the Cu loading density of Cu-N-C is approximately 16.37 wt%, thus the Cu-N-C@GF electrode can

provide more active sites for better reaction performance. In one word, our Cu-N-C@GF-based system not only exhibits a substantial $\Delta E_{\text{Cu-N-C}}$ but also offers a wealth of reaction sites, facilitating the efficient driving of catalytic reactions.

Figure R10-13 have been added into the revised manuscript as Figure S19, S22, S23 and Figure 2g, h, with a proper description.

Figure R11. (a) SEM image of CuO@GF. (b) SEM image of CuO. (c) The measured XRD pattern of CuO. (d) Photograph of the double-chamber reactor we set to test performance of CuO@GF. (e) The corresponding measured BPA removal performance with the set in d. Conditions: BPA = 5 μM , PDS = 10 mM, NaCl = 0.5 wt%. (f) The measured open-circuit potential curve on the CuO@GF. Conditions: BPA = 0.1 mM, PDS = 1 mM. Reference electrode: silver/silver chloride electrode (Ag/AgCl). Counter electrode: a carbon rod electrode. Working electrodes: CuO@GF electrode. (g) Measurement of current-time curves with different potentials on the CuO@GF electrode. Conditions: BPA = 0.1 mM, PDS = 1 mM. Reference electrode: silver/silver chloride electrode (Ag/AgCl). Counter electrode: a carbon rod electrode. Working electrodes: CuO@GF electrode.

Figure R12. (a) SEM image of CNTs@GF. (b) SEM image of CNTs. (c) TEM image of CNTs. (d)

Photograph of the double-chamber reactor we set to test performance of CNTs@GF. (e) The corresponding measured BPA removal performance with the set in d. Conditions: BPA = 5 μM , PDS = 10 mM, NaCl = 0.5 wt%. (f) The measured open-circuit potential curve on the CNTs@GF. Conditions: BPA = 0.1 mM, PDS = 1 mM. Reference electrode: silver/silver chloride electrode (Ag/AgCl). Counter electrode: a carbon rod electrode. Working electrodes: CNTs@GF electrode. (g) Measurement of current-time curves with different potentials on the CNTs@GF electrode. Conditions: BPA = 0.1 mM, PDS = 1 mM. Reference electrode: silver/silver chloride electrode (Ag/AgCl). Counter electrode: a carbon rod electrode. Working electrodes: CNTs@GF electrode.

Figure R13. (a) The removal ratio of BPA in the DOP towards different catalysts. Conditions: BPA = 5 μM , PDS = 10 mM, NaCl = 0.5 wt%. (b) Comparison of potential differences at different electrodes.

2) The authors used bisphenol A (BPA) as a model pollutant to study the decoupled AOP with a BPA concentration of 5 μM . A few other phenols were also studied. Would 5 μM of BPA be relevant to any contaminated water or wastewaters? The authors should focus on more “realistic” pollutants with environmentally relevant concentrations rather than the well-studied model BPA and phenols. For instance, chemical oxidation demand (COD) would be a more suitable parameter for real wastewaters.

Further, the authors mainly reported the removal of BPA, but with very limited focus on the end products. For complete pollutant degradation, mineralization rather than removal of the parent compound should be considered. What was the mineralization rate of BPA (how much BPA was fully mineralized under the experimental condition)? How did the toxicity of the end products compare to BPA?

Response: The reviewer’s comment has been well taken. Bisphenol A (BPA) is a representative endocrine disruptor which can cause adverse effects to humans and animals, including an observed rise in the incidence of breast and testicular cancers, diminished sperm counts, and premature onset of puberty (*Trends Endocrinol. Metabol.* **1998**, 9, 3, 124-128; *Endocrinology* **2009**, 150, 6, 2964-2973; *Chem. Res. Toxicol.* **2014**, 27, 9, 1463-1473). It is estimated that the global BPA market will reach a value of USD 35.59 billion by 2028 (*Chemosphere* **1998**, 36, 10, 2149-2173; *Research and Markets* **2021**, Global bisphenol A (BPA) market report and forecast 2021-2026). The abundant production and use of BPA will result in the discharge of BPA into the environment. And the detection of BPA in natural aquatic environments has been widely reported (*Water Res.* **2002**, 36, 6, 1429-1438; *Environ. Sci. Technol.* **2016**, 50, 16, 8403-8416). Thus, based on consideration of the above situation, we chose BPA as the model pollutant to

demonstrate the capability of our device for the organic contaminate treatment.

We further measured the chemical oxygen demand (COD) change of the treated BPA solution after a 2-hour reaction, in comparison with the initial solution (Figure R14 below). As can be seen, COD of the solution was efficiently decreased to approximately 20% of the original one, suggesting the excellent treatment property of our system. The degradation products of BPA have been analyzed and presented in Figure S7 of the original manuscript (Figure R15a below). We further analyzed toxicity of these products via the ecological structure-activity relationship model (ECOSAR), by evaluating risk of the organic compounds exposed to fish, daphnia, and green algae (Figure R15b and c below). It clearly shows that both the acute toxicity and chronic toxicity (ChV) of the products decreased along with the degradation path, suggesting that our system can help to reduce the toxicity of the organic compound efficiently.

Furthermore, we investigated the potential of our strategy towards other types of organic pollutants (Figure R16 below). As can be seen, our set only not show excellent removal efficiency towards phenolic compounds, but also to antibiotic compound sulfamethoxazole and heterocyclic compound furfuryl alcohol. The removal performance towards organic compound benzoic acid and p-nitrophenol is not good, we attribute this to the electro-withdrawing functional groups on the compounds to prevent the reactions (*Environ. Sci. Technol.* **2020**, 54, 13, 8464–8472; *Chem. Engin. J.* **2020**, 394, 124864; *Environ. Sci. Technol.* **2019**, 53, 24, 14595–14603). The good removal efficiency of our system towards these organic compounds suggests its applicability for treatment of electro-rich organic compound impaired water.

Figure R14-16 have been added into the revised manuscript as Figure S8, S9 and S19 with a proper description.

Figure R14. Comparison of the COD values before and after a 2-hour degradation of BPA. Conditions: catalyst = 0.15 g/L, BPA = 0.1 mM, PDS = 1 mM.

Figure R15. (a) Proposed degradation pathway of BPA based on the detected intermediate products. (b) and (c) The respective acute and chronic toxicity analysis of BPA and its corresponding degradation products with the ecological structure-activity relationship model (ECOSAR). Green, yellow, orange and red represent harmless, harmful, means toxic and highly toxic respectively. LC₅₀: lethal concentration 50; EC₅₀: concentration for 50% of maximal effect; ChV: chronic value.

Figure R16. Monitored concentration change of different organic pollutants using the double-chamber reactor. Conditions: pollutants = 5 μM.

3) The experiments were performed in DI water spiked with BPA, and the complexity of actual water and wastewaters was not considered. Common water constituents, particularly dissolved organic matter can affect the efficiency of the proposed DOP. In addition, would the interaction between dissolved organic matter and electrode result in the formation of any undesired byproducts that can have adverse effect on the ecosystem?

Response: We appreciate the critical comment by the reviewer. The influence of typical inorganic ions in natural water, different pH values (range from pH 3 to 11) as well as the presence of humic acid (HA, a representative natural organic matter in literatures, e.g. *Angew.*

Chem. Int. Ed. **2023**, e20230326; *ACS Catal.* **2022**, 12, 14954-14963; *Water Res.* **2023**, 241, 120151) on the BPA removal performance were measured and the results were summarized in Figure R17a-c below. As can be seen, regardless of these factors, the removal efficiency of BPA consistently remains around 100%, underscoring the robustness of our system.

What's more, to further explore the influence of natural organic matters (NOMs) on the reaction performance, we purchased commercial simulated NOMs (Upper Mississippi River NOM No.1R110N) and mixed it with the BPA solution to conduct the degradation experiment (Figure R17d). The system maintains a consistently high BPA removal efficiency close to 100%, irrespective of the presence of the simulated NOMs. Moreover, we extended our investigation to measure the chemical oxygen demand (COD) change of a prepared NOMs solution (5 ppm) during a 2-hour reaction employing our DOP system. Encouragingly, the COD value of the solution remained unchanged before and after the reaction, providing clear evidence that no reaction occurred between the NOMs and the catalytic electrode, therefore excluded the possible formation of any undesired byproducts that can have adverse effect on the ecosystem (Figure R17e).

Figure R17 has been added into the revised manuscript as Figure S20 with a proper description.

Figure R17. (a) The removal performance of BPA under the presence of different inorganic ions with the DOP system. Conditions: BPA = 5 μ M, inorganic ions = 20 mM. (b) The removal performance of BPA under different pH values. Conditions: BPA = 5 μ M. (c) Comparison of the removal performance of BPA with and without the presence of humic acid. Conditions: BPA = 5 μ M, humid acid = 5 ppm. (d) Comparison of the removal performance of BPA with and without the addition of commercial simulated NOMs (Upper Mississippi River NOM No.1R110N). Conditions: BPA = 5 μ M, NOMs = 5 ppm. (e) The measured COD change of NOMs dissolved aqueous solution before and after conducting the Cu-N-C@GF based DOP reaction. Conditions: NOMs = 5 ppm. Note that all the measurements were conducted after the treatment with Cu-N-C@GF based DOP reaction if without further notice.

4) The zebra fish toxicity study was not well designed. It is common sense that strong oxidants like peroxysulphate (PDS) would have detrimental effect on aquatic organisms and it is fully expected that direct addition of PDS would cause zebra fish death. The authors showed that the DOP did not cause the death of zebra fish, but that does not necessarily mean the DOP is safe for practical application. First, as specified in a previous comment, the authors should compare the toxicity of the end products versus the parent compound, not with the unrealistic conventional AOP process. Second, the long-term toxicity should also be considered in addition to the acute toxicity.

Response: The reviewer's comment has been well taken. Please refer to our response to Q2 above for the evaluation of the decreased toxicity of the degradation products.

Additionally, we conducted supplementary experiments to assess the environmental safety of our methodology within an aquatic context. Two separate tanks were utilized, each containing clean water and housing a population of 20 zebrafish. One of these tanks was equipped with our floating DOP device, while the other tank remained untreated as a control. Over a two-week observation period (Figure R18), we meticulously monitored zebrafish mortality rates. Notably, the presence of our DOP device did not result in any discernible increase in zebrafish mortality when compared to the control group.

Furthermore, we established two analogous tanks, however, we substituted the contents of the tanks with the BPA solution (5 μM), instead of clean water used previously (Figure R18b). Notably, the tank containing the BPA solution with the integrated floating device exhibited a comparatively higher fish survival rate than the tank with the BPA solution lacking the treatment device. We attribute the diminished fish survival rate in the former tank to the toxic effects induced by BPA, whereas in the latter tank, the BPA concentration was progressively reduced by the device, resulting in fewer fish fatalities.

Additionally, we tested and compared the Cu^{2+} ion in the solution before and after the 14-day continuous experiment in Figure R18b (Figure R19 below). The non-detectable Cu^{2+} ion concentration (both the values are far below the detection limit (5.54 ppb) of flame atomic absorption spectrometry) suggested the stability and safety of our electrode.

We hope the above analysis can help to clarify the reviewer's concerns to the safety of our strategy and device to the environment.

Figure R18-19 has been added into the revised manuscript as Figure S28 with a proper description.

Figure R18. Survival of zebrafish with/ without the device for two weeks (a) in clean water (b) in simulated BPA contaminated wastewater. Conditions (if required): BPA concentration in the 4 L tank: 5 μM , PDS solution stored in the tube: 10 mM.

Figure R19. The detected copper ion concentration in the tank in Figure R18b before and after the 14-day continuous treatment.

Reviewer #3(Remarks to the Author):

The authors reported a decoupled AOP system consisting of a floating device for the degradation of organic pollutants in a hazard-free manner. They evaluated the performance of a designed device by testing the degradation of various contaminants and identified the active site of the catalyst (Cu-N-C@GF) via computational analysis. They also conducted an environmental impact assessment of PDS (oxidant). Finally, they investigated the feasibility of the integrated decoupled AOP device for potential in-situ wastewater treatment and resource recovery. This research is very interesting and opens up a lot of possibilities for water treatment. However, this manuscript is difficult to accept in its current form and requires major revision. The detailed comments are as follows.

Response: We sincerely appreciate the reviewer for the acknowledge of the significance of our work. We have conducted more experiments to elaborate our work better, in accordance to the reviewer's comments. Additionally, the "decoupled AOP" has been rephrased to "decoupled oxidation process (DOP)" in our revised manuscript. Please refer the detailed point to point response below.

1) Why did the authors use the Cu-N-C@GF catalyst? This needs to be explained in the introduction.

Response: The reviewer's comment has been well taken. As we illustrated in the manuscript, generation of a positive potential difference (ΔE) between the cathodic and anodic ends is the prerequisite for the DOP reaction to occur. What's more, the number of reaction sites on the electrode also matters greatly to the reaction performance. In consideration of these, Cu-N-C was selected as the catalyst to demonstrate our DOP concept based on two merits of it. First, as we demonstrated in the manuscript, Cu-N-C catalyst can help to generate a relatively large potential difference ($\Delta E_{\text{Cu-N-C}} \approx 0.5$ V) between the organic compound and PDS ends, to effectively drive the electrons transfer process. Meanwhile, as we elaborated in the manuscript, the atomically dispersed CuN_2 sites on Cu-N-C mainly serve as the absorption site for PDS and BPA, as well as activation site for PDS. Since the Cu loading amount on Cu-N-C reached around 16.37 wt%, the Cu-N-C can provide sufficient reaction sites for the catalytic reaction to demonstrate our strategy.

To further manifest our design concept, we selected other commercial catalyst such as CuO and carbon nanotubes (CNTs) which can also create ΔE within the decoupled system. Figure R20 and R21 showed the characterization of the prepared CuO@GF and CNTs@GF electrode, respectively. With the prepared electrodes, similar degradation of BPA through a DOP process with the two electrodes were conducted, respectively. As can be seen, they both showed the degradation of BPA in a certain degree (Figure R20 and R21). As measured, positive ΔE and current were both detected in the two reaction sets, which support our strategy greatly.

It should be mentioned that, although composite CuO and CNTs can trigger the DOP, our Cu-N-C presents the best organic pollutant removal performance (Figure R22). We attributed this phenomenon to two main reasons, first is the larger $\Delta E_{\text{Cu-N-C}}$ (0.5 V) than ΔE_{CuO} (0.34 V) and ΔE_{CNTs} (0.23 V). Second reason is the abundant atomically dispersed Cu sites on the

electrode surface to catalyze the reaction. As we mentioned in the manuscript, the atomically dispersed Cu-N₂ served as the active sites for the catalytic reaction, and the Cu loading density of Cu-N-C is approximately 16.37 wt%, thus the Cu-N-C@GF electrode can provide more active sites for better reaction performance. In one word, our Cu-N-C@GF-based system not only exhibits a substantial $\Delta E_{\text{Cu-N-C}}$ but also offers a wealth of reaction sites, facilitating the efficient driving of catalytic reactions.

Figure R20-22 have been added into the revised manuscript as Figure S22, S23 and Figure 2g-h with a proper description. The third paragraph of the introduction section in the revised manuscript has also been revised to elaborate the advantage of our fabricated catalyst Cu-N-C as well as the versatility of our strategy, which was shown as below:

“...This approach relies on efficient catalyst capable to create sufficient positive potential difference (ΔE) between the oxidative and reductive two half-reactions within an AOP, thus triggering the decoupled two reactions. Single-atom catalysts have emerged as a promising material design option for various catalytic processes since efficiency of the atomic reaction sites can be maximized to nearly 100%.³²⁻³⁵ We identified a copper based single atom catalyst (Cu-N-C) with high catalytic activity to create a substantial potential difference ($\Delta E = 0.5$ V) between the oxidative and reductive two half-reactions. The high loading of atomically dispersed Cu atom sites (16.37 wt%) on the composite provides abundant reaction sites (absorption and catalytic sites) to facilitate catalytic reactions...” And “...This strategy can also be validated in other commercial catalysts such as CuO, carbon nanotubes...”.

Figure R20. (a) SEM image of CuO@GF. (b) SEM image of CuO. (c) The measured XRD pattern of CuO. (d) Photograph of the CuO@GF electrode based double-chamber reactor for degradation of BPA. (e) The measured BPA removal performance in accordance to d. (f) The measured open-circuit potential curve on CuO@GF. Reference electrode: silver/silver chloride electrode (Ag/AgCl). Counter electrode: a carbon rod electrode. Working electrodes: CuO@GF electrode. (g) Measurement of current-time curves with different potentials on the CuO@GF electrode. Reference electrode: silver/silver chloride electrode (Ag/AgCl). Counter electrode: a carbon rod electrode. Working electrodes: CuO@GF electrode. Conditions (if required): BPA = 0.1 mM, PDS = 10 mM, NaCl = 0.5 wt%.

Figure R21. (a) SEM image of CNTs@GF. (b) SEM image of CNTs. (c) The measured XRD pattern of CNTs. (d) Photograph of the CNTs@GF electrode based double-chamber reactor for degradation of BPA. (e) The measured BPA removal performance in accordance to d. (f) The measured open-circuit potential curve on the CNTs@GF. Reference electrode: silver/silver chloride electrode (Ag/AgCl). Counter electrode: a carbon rod electrode. Working electrodes: CNTs@GF electrode. (g) Measurement of current-time curves with different potentials on the CNTs@GF electrode. Reference electrode: silver/silver chloride electrode (Ag/AgCl). Counter electrode: a carbon rod electrode. Working electrodes: CNTs@GF electrode. Conditions (if required): BPA = 0.1 mM, PDS = 10 mM, NaCl = 0.5 wt%.

Figure R22. (a) Comparison of the BPA removal ratio with the employment of different catalytic electrodes. Conditions: BPA = 5 μM, PDS = 10 mM, NaCl = 0.5 wt%. (b) Comparison of potential differences at different electrodes.

2) Can the Cu-N-C@GF catalyst itself activate PDS? It is necessary to evaluate the PDS activation efficiency of the catalyst itself without using a decoupled AOP system. If PDS is activated on the catalyst, it is doubtful that a decoupled AOP system is necessary.

Response: We thank the reviewer for the thoughtful thinking. The catalyst Cu-N-C cannot activate PDS in the only existence of PDS, but it can trigger the activation of PDS in the co-existence of PDS and organic pollutants (Figure R23 below). As we mentioned in the manuscript, the catalytic reaction occurs only in the co-existence of organic pollutants and PDS. The potential difference between the two ends will drive the transferring of electrons from

organic compounds to PDS. The sole existence of Cu-N-C@GF cannot activate PDS directly, as we measured and presented in Figure S8c of the original manuscript (now is Figure S9c of the revised manuscript, which is also shown as Figure R23 below). Similar phenomenon was observed in the sole presence of BPA. The co-existence of BPA and Cu-N-C can efficiently cause change on PDS concentration.

Figure R23. (a) Comparison of the activation of PDS in the sole or co-presence of Cu-N-C@GF and BPA. Conditions: BPA = 0.1 mM, PDS = 1 mM.

Advantage of our DOP system can be elaborated through three aspects. Firstly, as we mentioned in the introduction section of the manuscript, in conventional AOPs process, a directional addition of oxidants into the contaminated water is a prerequisite, facilitating the generated active species to oxidize the pollutants. But a directional adding of oxidants into the treated water would put a threat to aquatic organisms and even cause death of them. As demonstrated in Figure 3e and f of our manuscript (shown as Figure R24 below), if the PDS directly added into the 4 L water containing 20 zebrafish, all the fish dead within 5 hours. On the contrary, with our decoupled system to storage the PDS in a physically separated tube above the water surface, almost all the fish stay alive. Therefore, a DOP system is necessary to avoid the oxidative damage or death of the aquatic lives in water caused by direct addition of PDS.

Secondly, as we mentioned in the manuscript, since the reaction only occurs at the co-existence of PDS and BPA, the device possesses a self-responsive characteristic that allows it to respond to changes in concentration of organic contaminants. To manifest that, we simulated an application of our method in a consecutive water treatment condition (Figure 3d of the original manuscript, which was presented as Figure R25 below). At the beginning, a device filled with 40.0 mL of PDS (10.0 mM) solution was floated on 2.0 L of clean water in a water tank. Before the addition of BPA into the tank, the concentration of PDS in the reactor was almost unchanged. Once BPA (5.0 μ M) was added into the water tank, the concentrations of BPA and PDS spontaneously decreased until the complete depletion of BPA. The device was capable in responding to the appearance of BPA anytime till the stored PDS was completely consumed. The degradation rate of the re-filled BPA decreased as the concentration of PDS decreased, due to the slowdown of the cathodic reaction in the reactor. The BPA removal rate could be recovered to its initial state, when the reactor was replenished with fresh PDS solution.

Thirdly, benefiting from the decoupled reduction of PDS in isolated reactors, it is feasible to use the modular devices for long-term on-site water treatment by just regularly replacing

PDS in reactors. Meanwhile, in stark contrast to conventional AOPs in which PDS was added directly into target water, the used PDS in our approach can be easily recovered without leaving any adverse by-products in treated water. The final product of PDS after the redox half-reaction is sulfate. We demonstrated that it was possible to fully recover sulfate from the used PDS by adding CaCl_2 . The collected CaSO_4 salt can be further used as raw material in many applications including fabrication of cement, agriculture fertilizer, additive for paper manufacturing, and so on (Figure 4d of the original manuscript, which is shown as Figure R26 below), indicating a sustainable water treatment strategy with less footprint.

Figure R24b has been added into the revised manuscript as Figure S27, with a proper description.

Figure R24. (a) Photographs taken at different time of the zebrafish in the water tank during the BPA degradation experiment, with the DOP and the conventional AOP, respectively. (b) The monitored mortality rate of zebrafish in different situation. For the four groups from left to the right, tank with BPA-contaminated wastewater, tank with BPA-contaminated wastewater equipped with a floating treatment device, tank with BPA-contaminated wastewater supplemented with PDS, and tank with BPA-contaminated wastewater supplemented with both PDS and a floating treatment device. Conditions (if required): BPA = 5 μM , PDS = 1 mM (in the tank)/10 mM (40 mL, in the tube), NaCl = 0.5 wt%. The error bars in f were based on parallel experiments of three groups of zebrafish. NS refers to no significance. *** $p < 0.001$, ** $p < 0.01$, or * $p < 0.05$.

Figure R25. Monitoring of the concentration change of PDS and BPA during the designed consecutive reaction. Conditions: BPA = 5 μM , PDS = 10 mM, NaCl = 0.5 wt%.

Figure R26. The cycled consumption and replenishment of PDS during the reaction, and recovery of CaSO_4 from the used PDS waste liquid for other potential industrial use.

3) Did the author test BPA degradation in a reactor without PEM? I am not sure if PEM is really necessary in this system.

Response: We thank the reviewer for the thoughtful thinking of our device. The Cu-N-C catalyst can catalyze the oxidation of BPA with the co-existence of PDS in a single chamber reactor. The catalytic performance of Cu-N-C was evaluated in Figure R27a for degradation of BPA via activating PDS. As can be seen, the sole presence of PDS didn't cause obvious change on BPA concentration, while a complete BPA removal was achieved with the co-existence of PDS and Cu-N-C. This is also driven by the potential difference created between the PDS and BPA, under the assistance of Cu-N-C (Figure R27b). A possible reaction pathway on the powder catalyst Cu-N-C was schematically presented in Figure R27c. Once BPA and PDS met Cu-N-C, they will be adsorbed on Cu-N-C and formed a positive ΔE between the adsorbed PDS* and BPA*. The positive ΔE will drive the electrons to transfer from the BPA* to PDS* ends, resulting in the oxidation of BPA. The intrinsic potential difference driven redox reaction thus can be spatially divided into two half reactions in two chamber, facilitating the DOP (Figure R28).

As the schematic diagram shown in Figure 1a of our manuscript (which is shown as Figure R29 below), utilization of the proton exchange membrane (PEM) is to spatially separate the reduction of PDS and oxidation of BPA, and to block the exchange of solvents/reactants while allowing the migration of small cations (e.g. H^+ , Na^+ ions) between the two chambers to maintain the charge neutrality during reactions. Thus, the PEM is necessary for the DOP. The advantages of the DOP has been illustrated in response to Q2 above.

Figure R27 and R28 have been added into the revised manuscript as Figure S12 and S13, with a proper description.

Figure R27. (a) The monitored BPA concentration change under different condition. Conditions (if required): catalyst = 0.15 g/L, BPA = 0.1 mM, PDS = 1 mM. (b) The Open-circuit potential curve measurement of BPA (0.1 mM) and PDS (1 mM) solutions with different triple electrode system. Reference electrode: silver/silver chloride electrode (Ag/AgCl). Counter electrode: platinum electrode. Working electrodes: glassy carbon electrode, and glassy carbon electrode coated with Cu-N-C. (c) Scheme illustration of the electron transfer process involved in the oxidation of BPA under the assistance of powder Cu-N-C catalyst and PDS.

Figure R28. Schematic diagram of design ideas from the catalyst powder to the device.

Figure R29. Schematic diagram for the DOP conducted in a double-chamber reactor, with interconnection part sealed with PEM.

4) Why was NaCl (0.5 wt%) added into the reactor? What is the role of NaCl? Is it possible reactive chlorine species to participate in the BPA degradation reaction? I don't understand why NaCl was added.

Response: We thank the reviewer for the careful thinking of our manuscript. We introduced a minimal quantity of NaCl at a concentration of 0.5 wt% as an electrolyte to enhance ion migration in the solution, thereby facilitating the DOP. Our investigation also examined the impact of varying NaCl concentrations on the degradation experiment. As illustrated in Figure R30a (Supplementary Figure 16 of original manuscript, now as Supplementary Figure 27b of the revised version), a minor addition of NaCl (0.5 wt%) suffices to support cation transfer for charge neutrality maintenance. Consequently, we selected a 0.5 wt% NaCl concentration, which falls within the intermediate range of NaCl concentrations typically found in surface freshwater (NaCl: 0~1 wt%). (*Sci. Data* **2020**, 7, 231; *Heliyon* **2023**, 9, e18685).

To explore the possibility of generation of $\text{Cl}\cdot$ radical in our system, *tert*-butyl alcohol (TBA, a typical scavenger for $\text{Cl}\cdot$, reaction rate with $\text{Cl}\cdot$: $3\text{-}19 \times 10^8 \text{ M}^{-1} \text{ s}^{-1}$) was added during

the degradation experiment (*Environ. Sci. Technol.* **2023**, 57, 9416-9425; *Water Res.* **2016**, 90, 15-23; *Environ. Sci. Technol.* **2016**, 50, 18, 10143-10152). As unveiled by Figure R30b below, addition of TBA didn't cause obvious change on the BPA degradation performance, thus we can conclude that Cl^- didn't participate in our reaction.

Figure R30b have added to the revised manuscript as Supplementary Figure 29b with a proper description.

Figure R30. (a) Measurement of the degradation curve of BPA under different concentration of NaCl. (b) The removal performance of BPA with/without the presence of *tert*-butyl alcohol. Conditions (if required): BPA = 5 μM , PDS = 10 mM, NaCl = 0.5 wt%, TBA = 0.5 mM.

5) This is a similar question to comment 2. In Figure 3e, if PDS is added to a tank with a device, can PDS be catalytically activated? If this happens, the authors should compare the mortality rate of zebra fish when PDS is added in a tank with a modular device. While the authors have confirmed that PDS has an adverse effect to zebra fish, it remains to be seen how PDS affects zebrafish when it is activated by the catalyst.

Response: As we respond to Q2 and Q3 above, the PDS can be activated by Cu-N-C in the co-existence of BPA. Thus, the direct addition of PDS into the tank with an electrode can activate the PDS, in the co-presence of BPA in the tank. However, direct contact of PDS with zebrafish results in significant harm and mortality, even in the presence of a catalyst. To provide a clearer illustration of this point, we compared zebrafish mortality rates when PDS was introduced directly into a tank versus when it was separately stored in the upper tube of our device. Each group test involved 20 zebrafish. As depicted in Figure R31, we monitored zebrafish mortality under four conditions: simulated BPA wastewater with zebrafish, simulated BPA wastewater with zebrafish and one floating device, simulated BPA wastewater with zebrafish and the addition of PDS (1 mM), and simulated BPA wastewater with zebrafish, the addition of PDS (1 mM), and a floating device.

In the fourth group, the presence of a catalytic electrode in the tank led to a slower decline in zebrafish vitality, attributed to the catalytic decomposition of PDS, compared to the third group. However, direct addition of PDS into the tank, whether with or without our device, resulted in the demise of all zebrafish within 5 hours. Conversely, when PDS was stored separately in the upper tube (the second group), fish mortality was negligible. Consequently, direct PDS contact with zebrafish, even with a catalyst present, induces harm and fatality. Thus, a DOP system is imperative to prevent harm to aquatic life coexisting with conventional AOPs.

Figure R31 have been added into the revised manuscript as Figure S26, with a proper description.

Figure R31. The monitored mortality rate of zebrafish under four different conditions. From left to the right, the simulated BPA wastewater with zebrafish, the simulated BPA wastewater with zebrafish and the presence of the floating device, the simulated BPA wastewater with zebrafish and direct addition of PDS (1 mM) into the water, the simulated BPA wastewater with zebrafish, addition of PDS (1 mM) and the presence of the floating device. Conditions: BPA = 5 μ M, NaCl = 0.5 wt%, PDS = 1 mM (in the tank)/10 mM (40 mL in the tube), NaCl = 0.5 wt%. The error bars in f were based on parallel experiments of three groups of zebrafish. NS refers to no signification. *** $p < 0.001$, ** $p < 0.01$, or * $p < 0.05$.

6) When the catalyst was used for a long time or repeatedly, did the authors check for possible leaching of Cu ions from the catalyst? If leaching occurs, the authors need to confirm that leaching affects zebra fish.

Response: We conducted supplementary experiments to assess the environmental safety of our methodology within an aquatic context. Two separate tanks were utilized, each containing clean water and housing a population of 20 zebrafish. One of these tanks was equipped with our floating DOP device, while the other tank remained untreated as a control. Over a two-week observation period (Figure R32), we meticulously monitored zebrafish mortality rates. Notably, the presence of DOP device did not result in any discernible increase in zebrafish mortality when compared to the control group.

Furthermore, we established two analogous tanks, however, we substituted the contents of the tanks with the BPA solution (5 μ M), instead of clean water used previously (Figure R32b). Notably, the tank containing the BPA solution with the integrated floating device exhibited a comparatively higher fish survival rate than the tank with the BPA solution lacking the treatment device. We attribute the diminished fish survival rate in the former tank to the toxic effects induced by BPA, whereas in the latter tank, the BPA concentration was progressively reduced by the device, resulting in fewer fish fatalities.

Additionally, we tested and compared the Cu²⁺ ion in the solution before and after the 14-day continuous experiment in Figure R32b (Figure R33 below). The non-detectable Cu²⁺ ion concentration (both the values are far below the detection limit (5.54 ppb) of flame atomic absorption spectrometry) suggested the stability and safety of our electrode.

Figure R32 and R33 have been added into the revised manuscript as Figure S28, with a proper description

Figure R32. Survival of zebrafish with/ without the device for two weeks (a) in clean water (b) in simulated BPA contaminated wastewater. Conditions (if required): BPA concentration in the 4 L tank: 5 μM , PDS solution stored in the tube: 10 mM.

Figure R33. The detected copper ion concentration in the tank in Figure R32b before and after the 14-day continuous treatment.

7) Did the authors have a specific reason for using low concentration of BPA (5 μM) and high concentration of PDS (10 mM)? Did the authors use such extreme concentrations of BPA and PDS because of the low efficiency of the catalyst?

Response: Actually, different BPA concentration was used in our experiment for different purpose. At the beginning, to demonstrate the efficiency of our set, especially in comparison with literatures, a BPA concentration of 0.1 mM was adopted in the single chamber degradation experiment (Figure R34a below), which presented a fast removal of BPA. In comparison with literatures, the high k value manifested the excellent catalytic performance of our catalyst (Table R5). Furthermore, BPA solution with concentration of 5 μM was adopted in the evaluation of the safety of our DOP approach for impaired water with co-living aquatic organism. There are two main reasons for us to select 5 μM BPA solution for this experiment. First is that a high concentration of BPA will cause immediate death of the zebrafish, which will influence us to evaluate the effect of direct addition of PDS into the solution. Second is the currently detected BPA concentration in actual natural water is in range of 0-10 $\mu\text{g/L}$, so we set the value at 5 $\mu\text{g/L}$, to simulate the actual concentration detected in actual environment (*Toxicol. Environ. Chem.* **2013**, 95, 386-399; *Environ. Sci. Pollut. Res.* **2021**, 28, 23, 30242-30254).

It is noteworthy that the reason we use a relatively high concentration of PDS in the upper tube of our floating device is based on the consideration of the limited volume of the PDS solution (40 mL). The actual PDS dosage applied was 108 mg (40 mL \times 10 mM \times 270.32 g/mol).

In contrast, typical literature-based treatments of 2 L impaired water using conventional AOPs utilize a PDS concentration of 1 mM (*Environ. Sci. Technol.* **2021**, 55, 2110-2120; *Water Res.* 2019, 160, 405-414). This results in a calculated total PDS quantity of approximately 540 mg ($2\text{ L} \times 1\text{ mM} \times 270.32\text{ g/mol}$), which is significantly higher than the 108 mg of PDS used in our decoupled system.

Figure R34a, b and Table R5 have been added into the revised manuscript as Figure S12a, S17b and Table S4, with a proper description.

Figure R34. (a) BPA removal curve in different system. Conditions: catalyst = 0.15 g/L, BPA = 0.1 mM, PDS = 1 mM. (b) The removal performance of the DOP towards different concentration of BPA. Conditions: PDS = 10 mM, NaCl = 0.5 wt%.

Table R5. Comparison of the catalytic performance of Cu-N-C-8 with the reported literatures

Catalyst	Oxidant dosage	Pollutant (mg L ⁻¹)	Removal efficiency	k value (min ⁻¹)	TOF (L g ⁻¹ min ⁻¹)	Reference
Cu-N-C-8 (0.15 g/L)	PDS (1 mM)	BPA (0.1 mM)	100% (5 min)	1.777	11.85	This work
N-BC900 (0.2 g/L)	PDS (2 mM)	BPA (88 μM)	100% (20 min)	0.214	1.07	Environ. Sci. Technol. 2018, 52, 15, 8649
CNF3 (0.10 g/L)	PMS (1 mM)	4-CP (0.1 mM)	100% (20 min)	0.254	2.54	Environ. Sci. Technol. 2018, 52, 4, 2197
Fe-SACs (0.2 g/L)	PMS (1.3 mM)	BPA (0.11 mM)	88% (30 min)	0.104	0.52	Angew. Chem. Int. Ed. 2021, 60, 22513
Fe _{0.15} Mn _{0.85} O ₂ (0.04 g/L)	PMS (0.5 mM)	BPA (5 μM)	100% (21 min)	~0.23	~5.75	Environ. Sci. Technol. 2019, 53, 21, 12610
CuO (0.2 g/L)	PDS (40 μM)	2,4-DCP (5 μM)	100% (40 min)	0.062	0.31	Environ. Sci. Technol. 2014, 48, 10, 5868
NBC-4 (1.0 g/L)	PDS (1 mM)	SDZ (10 μM)	96.2% (60 min)	0.0798	0.0798	Water Res. 2019, 160, 405
Cu-N ₄ /C-B (0.1 g/L)	PMS (0.65 mM)	BPA (88 μM)	98% (5 min)	0.56	5.6	Proc. Natl. Acad. Sci. USA 2022, 119, e2119492119
PSS-800 (0.2 g/L)	PDS (1.85 mM)	2,4-DCP (0.62 mM)	100% (60 min)	0.0606	0.303	Appl. Catal. B: Environ. 2020, 260, 118160
Cu _{SA} -NC (0.04 g/L)	PDS (0.5 mM)	2,4-DCP (0.1 mM)	100% (30 min)	0.284	7.1	Environ. Sci. Technol. 2022, 56, 12, 8765
ZnFeMnO ₄ (0.1 g/L)	PMS (0.16 mM)	BPA (40 μM)	100% (15 min)	0.43	4.3	Proc. Natl. Acad. Sci. USA 2020, 117, 30966

Catalyst	Oxidant dosage	Pollutant (mg L ⁻¹)	Removal efficiency	k value (min ⁻¹)	TOF (L g ⁻¹ min ⁻¹)	Reference
Annealed CNT (0.1 g/L)	PDS (1 mM)	Phenol (0.1 mM)	100% (30 min)	0.198	1.98	Environ. Sci. Technol. 2020, 54, 1267

Note: The turnover frequency (TOF) was calculated through dividing the reaction rate of pollutant degradation by the catalyst concentration.

Table R6. Comparison of the dosage of PDS activated in different systems to degrade BPA

Methods	V _{BPA} ^a	C _{PDS} ^b	V _{PDS} ^c	D _{PDS} ^d
Traditional AOP	2 L	1 mM	2 L	540 mg
Our floating device with DOP process	2 L	10 mM	40 mL	108 mg

Note: a. The volume of BPA solution in different systems.

b. The concentration of PDS in different systems.

c. The volume of PDS solution in different systems.

d. The dosage of PDS in different systems. $D_{PDS} = C_{PDS} \times V_{PDS} \times 270.32$ (Formula weight of PDS).

REVIEWER COMMENTS

Reviewer #1 (Remarks to the Author):

The authors have augmented the data presentation supporting the mechanisms in which the decouple oxidation process operates, and up-scaled the proof-of-concept study with a 1-m pool of water. Despite the effort, the fundamental questions - what scenarios would make such floating device a compelling option for water purification considering the volume of water, the size of the water bodies, and costs associated with applying such option. This comment does not necessarily pose as criticism, but the authors will have to identify scenarios, in water purification purposes or beyond, where the floating device might find a niche application.

Reviewer #2 (Remarks to the Author):

This reviewer acknowledges the extensive efforts that the authors have taken to address the reviewer's comments. The quality of the manuscript has been improved. Meanwhile, several aspects still need further clarification/justification/revision, as specified below.

1. The authors underestimate the cost to fabricate their electrodes and floating devices. The fabrication cost presented in Table R4 of the Response to Reviews only considered the material cost, but not the cost of the synthesis process. Given the high complexity of the Cu-N-C@GF electrode and related precursors, the actual fabrication cost is likely way higher than the authors estimated. The scalability of the complex preparation process of the Cu-N-C@GF electrode is also of question. Similarly, the total cost of the floating device in Table R1 of the Response to Reviews (Supplementary Table 5 in the Supplementary Information) is also underestimated due to the underestimation of the electrode fabrication cost. The actual cost would further increase, considering the long-term use/stability of the device.

2. The authors used 5 μM of BPA and a few other model pollutants to evaluate the performance of their DOP process. However, the concentrations were likely orders of magnitude higher than those observed in real water/wastewater systems. The performance reported in this manuscript therefore cannot be directly applicable to real water/wastewater systems, and this limitation should be clearly acknowledged.

3. The proposed DOP concept is nice. However, this reviewer still does not believe the necessity to fabricate a highly complex electrode consisting of single-atom Cu to achieve this, other than eye-catching. As the authors showed in their revised manuscript, the DOP could be achieved with the use of much simpler electrode, despite the slightly reduced performance.

Reviewer #3 (Remarks to the Author):

This manuscript has been well revised and highly improved based on the reviewer's comments. It seems to be well explained, especially as it relates to cost and practicality of this system. There are some minor points that should be addressed before this manuscript can be accepted.

1. In Scheme 1 (for DOP), the authors represented that the contaminants were fully mineralized (i.e., the complete conversion of contaminants to H₂O and CO₂). However, no results related to mineralization are found in the manuscript. Therefore, the authors should provide total organic carbon (TOC) removal data.
2. The performance of the integrated modular device needs to be validated in real wastewater. Once this is addressed, the practicality of this device will be more apparent.

Responses to the reviewers' comments (Manuscript number NCOMMS-23-14745A-Z)

Changes in the revised manuscript as a response to the reviewers' comments are highlighted in red color and clarifications regarding the reviewer's comments are provided in blue color.

Reviewer #1 (Remarks to the Author):

The authors have augmented the data presentation supporting the mechanisms in which the decouple oxidation process operates, and up-scaled the proof-of-concept study with a 1-m pool of water. Despite the effort, the fundamental questions - what scenarios would make such floating device a compelling option for water purification considering the volume of water, the size of the water bodies, and costs associated with applying such option. This comment does not necessarily pose as criticism, but the authors will have to identify scenarios, in water purification purposes or beyond, where the floating device might find a niche application.

Response: We appreciate the thoughtful review and the recognition of our research's contributions toward understanding the mechanism and executing a scaled-up proof-of-concept study of our decoupled oxidation process. The reviewer's insights have been invaluable in shaping our work.

We would like to address the advantages of the decoupled oxidation process (DOP) and the associated floating device in the following aspects:

- **In-situ organic pollutant remediation.** Our approach allows for the in-situ chemical breakdown of organic pollutants in impaired water without causing harm to aquatic organisms.
- **Self-responsive treatment.** The DOP approach embodies a self-responsive method for treating organic pollutants, triggering chemical reactions only in response to water contamination. This is vividly illustrated in Figure 3d of our manuscript, emphasizing the exclusive utilization of the oxidant PDS in the presence of BPA. This unique feature enables the convenient storage of the oxidant solution on the water surface as a proactive measure.
- **Effective treatment of micropollutants.** Our technique is especially effective for treating micropollutants at low concentrations (as depicted in Figure R1), as the reaction initiates upon the co-contact of PDS and organic compounds with the catalysts, as demonstrated in Figure 3d of the manuscript.

Regarding considerations about conventional off-site approaches, which typically rely on water pumps and may pose challenges in remote or isolated areas with limited power supplies and infrastructure, we've outlined potential water bodies where our in-situ floating device might find applicability:

- a. Rural water sources in remote or off-grid areas, such as rural springs, local ponds or reservoirs
- b. Industrial runoff areas, for example, contaminated ponds near industrial sites
- c. Natural water reservoirs, such as small ponds and wetlands.

We have duly incorporated a comprehensive discussion regarding the scenarios where our floating device could be effectively utilized in the "Discussion" section of the revised manuscript.

“The unique features of the DOP approach render it an ideal candidate for in-situ water treatment in remote or isolated regions where centralized water treatment is not feasible. Notably, this is applicable to rural water sources, encompassing rural springs, local ponds, or reservoirs. Moreover, our approach, characterized by its portability and

absence of environmental hazard release, exhibits significant potential in addressing accidental pollution events. This holds particular relevance in the context of natural water reservoirs, such as small ponds and wetlands, which serve as critical habitats for aquatic organisms. The self-responsive feature of our approach allows for the exclusive utilization of oxidants, conveniently stored on the water surface as a proactive and preventive measure. This introduces a novel, effective, and environmentally-friendly in-situ water treatment option, particularly suited for remote or off-grid areas cohabited by aquatic organisms.”

Figure R1. (a) Measured BPA removal performance with the floating device, towards 2 L water with different BPA concentrations. (b) Calculated removal ratio of pollutants with the floating device after 3 h of reaction time. Note that the organic mixture was a 2 L solution of water containing 2 µg/L BPA, 2 µg/L phenol, 2 µg/L *p*-chlorophenol and 2 µg/L 2,4-dichlorophenol. Conditions: PDS = 10 mM, NaCl = 0.5 wt%.

Reviewer #2 (Remarks to the Author):

This reviewer acknowledges the extensive efforts that the authors have taken to address the reviewer's comments. The quality of the manuscript has been improved. Meanwhile, several aspects still need further clarification/justification/revision, as specified below.

Response: We are deeply grateful to the reviewer for recognizing the improvements made in the revised manuscript. In response to supplementary comments, we have conducted additional experiments and provided further discussions. Please find below a detailed point-by-point response to elaborate on these enhancements.

1) The authors underestimate the cost to fabricate their electrodes and floating devices. The fabrication cost presented in Table R4 of the Response to Reviews only considered the material cost, but not the cost of the synthesis process. Given the high complexity of the Cu-N-C@GF electrode and related precursors, the actual fabrication cost is likely way higher than the authors estimated. The scalability of the complex preparation process of the Cu-N-C@GF electrode is also of question. Similarly, the total cost of the floating device in Table R1 of the Response to Reviews (Supplementary Table 5 in the Supplementary Information) is also underestimated due to the underestimation of the electrode fabrication cost. The actual cost would further increase, considering the long-term use/stability of the device.

Response: We appreciate the thoughtful insights provided by the reviewer. Tables R1 and R4 have been updated to include two additional cost factors: electricity consumed during catalyst fabrication and electrode stability. We've shown sustained performance of our electrodes over ten cycles, as depicted in Figure S21 of the manuscript, and considered the associated cost depletion in estimating treatment costs for 2500 m³ of impaired water.

Regarding the valid concern about potential fabrication cost disparities, especially on a large scale and for prolonged practical use, we aim to highlight that our cost tables underscore the current acceptability of costs at this developmental stage. Future scale-up endeavors will focus on optimizing the fabrication process for cost efficiency.

Our manuscript's core objective is to demonstrate the decoupled oxidation process (DOP) strategy and its advancements in water treatment, particularly compared to conventional advanced oxidation processes. The innovation lies in the conceptual framework and potential applications of the technology. We acknowledge that cost reduction can be achieved through ongoing optimization of catalyst synthesis and electrode fabrication.

To further support the general applicability of our DOP strategy, we have demonstrated that electrodes made from other commercial catalysts, such as CNTs and CuO, exhibit similar DOP functions. The history of CNTs, from its initial development in the 1990s (*Nature* **1991**, 354, 56; *Nature* **1993**, 363, 603) to the present, has witnessed many breakthroughs in mass production, significantly reducing prices from ~\$45,000 to ~\$100 per kilogram (*Small* **2013**, 9, 8, 1237). We're actively exploring strategies to employ more cost-effective catalysts and optimize device structures for enhanced performance. Through these continuous efforts, we're confident that the price of our electrode can be further reduced. We hope this rationale aligns with the reviewer's concerns and underscores our commitment to making this technology more accessible and cost-effective in the future.

Table R1 and R2 have been updated into the revised manuscript as Table S5 and S6 with a proper description of the estimated cost.

Table R1. The respective calculation of the fabrication cost for different electrode

Electrode	Resource	Unit Price ^{1,2} (\$)	Amount/size used for one electrode	Partial cost for one electrode (\$)	
Cu-N-C@GF	Material	Copper (II) acetate monohydrate	4.75/kg	5.08 mg	2.41×10^{-5}
		L-glutamic acid	1.40/kg	1.87 mg	2.62×10^{-6}
		1,3,5-benzenetricarboxylic acid	4.89/kg	2.81 mg	1.37×10^{-5}
		Dicyandiamide	2.79/kg	41.15 mg	1.15×10^{-4}
		Cu-N-C	43.10/kg	3.60 mg	1.55×10^{-4}
		Nafion solution	1.61/ mL	3.60 μ L	5.80×10^{-3}
		Graphite felts	21.00/m ²	6.00 cm ²	1.26×10^{-2}
Electricity	Fabrication of Cu-N-C	0.140/kWh	3.56×10^{-2} kWh	4.98×10^{-3}	
	Fabrication of Cu-N-C@GF	0.140/kWh	0.363 kWh	5.08×10^{-2}	
Estimated cost for one Cu-N-C@GF electrode				$\sim \\$ 0.0744$	
CNTs@GF	Material	CNTs	100.00/kg	3.60 mg	3.60×10^{-4}
		Nafion solution	1.61/ mL	3.60 μ L	5.80×10^{-3}
		Graphite felts	21.0/m ²	6.00 cm ²	1.26×10^{-2}
	Electricity	Fabrication of CNTs	-	-	-
		Fabrication of CNTs@GF	0.140/kWh	0.363 kWh	5.08×10^{-2}
Estimated cost for one CNTs@GF electrode				$\sim \\$ 0.0696$	
CuO@GF	Material	CuO	11.32/kg	3.60 mg	4.08×10^{-5}
		Nafion solution	1.61/ mL	3.60 μ L	5.80×10^{-3}
		Graphite felts	21.0/m ²	6.00 cm ²	1.26×10^{-2}
	Electricity	Fabrication of CuO	-	-	-
		Fabrication of CuO@GF	0.140/kWh	0.363 kWh	5.08×10^{-2}
Estimated cost for one CuO@GF electrode				$\sim \\$ 0.0692$	

Note: ¹Prices of the chemicals were determined based on the average prices on website. ²Price of the electricity was determined according to the average electricity price in China.

Table R2. Comparison of estimated cost for treating 2500 m³ amount of impaired water with different techniques

	Construction cost for 2500 m ³ water treatment ability	Operation cost			Characteristics of operation	Total Cost/2500 m ³ impaired water
		Electricity consumption /m ³ impaired water	Cost depletions	Chemical usage/2500 m ³ impaired water		
Our floating device with decoupled oxidation process (DOP)	Electrodes + Floating set $\sim \$ 5.9$ per set $\$ 5.9 \times 100$ sets = $\\$ 590$ for treating 2500 m ³ impaired water at site ^a	No power consumption	Cu-N-C@GF electrode $\sim \$ 0.73$ per set $\$ 0.73 \times 100$ sets $\div 10 =$ $\\$ 7.3^d$	~ 2.7 kg of PDS, $\sim \$ 2.7^e$	in-situ and “self-responsive”	$\sim \\$ 600$

Conventional AOP water treatment station	Water tank+		No loss of equipment in short term	~2.7 kg of PDS, ~\$ 2.7 ^f	off-site, need monitoring on the occurrence of pollutants	>\$ 13,000
	Pump +	~3.12				
	Pipeline +	kWhm ⁻³ ×				
	Electric	~\$ 0.36				
	generator +	kWh ⁻¹ ×				
Treatment	2500					
system	m ³ =\$ 2808 ^c					
	>\$ 10,000^b					

Note: a. ¹The set used here is the enlarged set which can contain 1 L 10 mM PDS solution. ²The price was calculated based on the materials involved in the device fabrication, including the graphite felts, chemicals for catalyst synthesis, cylinder tube, copper wire and etc.

b. Prices of the equipment were determined based on the average prices on website.

c. The average kWhm⁻³ was calculated based on the average amount given in references (*Sci. Total Environ.* **2021**, 782, 146599).

d. According to Figure S21 of the original manuscript, our catalytic electrode was stably for 10-time degradation, thus the loss coefficient of the catalytic electrode was estimated to be 0.1.

e. The consumed PDS amount and price was calculated based on the experiment on Figure R1, Table R1 and price searched on website.

f. We estimated the PDS amount in conventional AOP as the same amount in our decoupled AOP, however, in dealing with trace micropollutants, the conventional AOPs always require oxidants with an overdose amount, due to the radical scavenging effects (*Environ. Sci. Technol.* **2021**, 55, 14494-14514; *Nature* **2016**, 529, 190-194).

2) The authors used 5 µM of BPA and a few other model pollutants to evaluate the performance of their DOP process. However, the concentrations were likely orders of magnitude higher than those observed in real water/wastewater systems. The performance reported in this manuscript therefore cannot be directly applicable to real water/wastewater systems, and this limitation should be clearly acknowledged.

Response: The reviewer's constructive feedback has led us to conduct additional experiments aimed at enhancing the applicability of our study to real-world scenarios. We've utilized lower concentrations of organic compounds to evaluate our floating device's performance, aligning with detected Bisphenol A (BPA) concentrations reported in certain surface waters, which can reach up to 20 µg/L (*Environ. Toxicol. Pharmacol.* **2015**, 40, 241). Considering the U.S. Environmental Protection Agency (EPA) predicted no-effect concentration (PNEC) for BPA at 0.175 µg/L in their action plan (2010), we assessed concentrations of 20, 2, and 0.2 µg/L to comprehensively evaluate treatment performance.

The results, depicted in Figure R2, illustrate the consistent removal efficacy of the floating device across all three concentrations, including when challenged with a mixture of organic compounds such as BPA, phenol, p-CP, and 2,4-DCP. Although removal rates may vary, it's noteworthy that the overall removal ratios for each compound reached 100%. To supplement this data, we've incorporated Figure R2 into the revised manuscript as Figure S29, accompanied by a detailed description.

We agree with the reviewer that the performance of our device may vary concerning different concentrations and organic compounds, and we have added corresponding description into the revised manuscript:

“However, in practical application, the removal rate may vary depending on the type and concentration of organic pollutants.”

Figure R2. (a) Measured BPA removal performance with the floating device, towards 2 L water with different BPA concentrations. (b) Calculated removal ratio of pollutants with the floating device after 3 h of reaction time. Note that the organic mixture was a 2 L solution of water containing 2 µg/L BPA, 2 µg/L phenol, 2 µg/L *p*-chlorophenol and 2 µg/L 2,4-dichlorophenol. Conditions: PDS = 10 mM, NaCl = 0.5 wt%.

3) The proposed DOP concept is nice. However, this reviewer still does not believe the necessity to fabricate a highly complex electrode consisting of single-atom Cu to achieve this, other than eye-catching. As the authors showed in their revised manuscript, the DOP could be achieved with the use of much simpler electrode, despite the slightly reduced performance.

Response: We express our gratitude to the reviewer for recognizing the value of the DOP strategy. We agree that the cost of our single-atom Cu-based electrode, Cu-N-C@GF, may surpasses those employing commercialized catalysts, such as CNTs@GF and CuO@GF. However, our deliberate choice to use the single-atom Cu catalyst stems from its substantial potential difference generated ($\Delta E_{\text{Cu-N-C}} \sim 0.5 \text{ V}$) and the abundant exposed reaction sites (CuN_2), both contributing to more efficient catalytic performance.

As demonstrated in the revised manuscript, electrodes employing more economical yet commercially available catalysts, namely CuO and CNTs, exhibit analogous functionalities, albeit with relatively lower efficiency, reinforcing the generality of our proposed strategy (see Figure R3). Moreover, we are currently designing an all-carbon-based, all-in-one electrode prepared in a single step, demonstrating comparable performance, as preliminary data showcased in Figure R4.

Regarding the historical development of CNTs since the 1990s, the substantial reduction in cost from approximately ~\$45,000 to the current ~\$100 per kilogram is a noteworthy point. This trajectory indicates that with advancements in materials synthesis and device fabrication techniques, the cost of the proposed device could foreseeably be significantly reduced in the near future. We remain optimistic about the potential cost efficiencies that ongoing advancements in materials science and fabrication techniques may bring.

Figure R3. (a) Comparison of the BPA removal ratio with the employment of different catalytic electrodes under conditions: BPA = 5.0 μ M; PDS = 10.0 mM; NaCl = 0.5 wt%. (b) Comparison of potential differences at different electrodes.

Figure R4. (a) Photograph of the carbon based catalytic electrode we prepared based on a one-step synthesis technique. (b) The BPA removal performance measured with the prepared carbon-based electrode base DOP double-chamber reactor under conditions: BPA = 5.0 μ M; PDS = 10.0 mM; NaCl = 0.5 wt%.

Reviewer #3 (Remarks to the Author):

This manuscript has been well revised and highly improved based on the reviewer's comments. It seems to be well explained, especially as it relates to cost and practicality of this system. There are some minor points that should be addressed before this manuscript can be accepted.

Response: We extend our sincere appreciation to the reviewer for recognizing the efforts taken to enhance the quality of our work. We have performed more experiments according to the reviewer's comments and revised our manuscript accordingly. Please refer to the detailed response below.

1) In Scheme 1 (for DOP), the authors represented that the contaminants were fully mineralized (i.e., the complete conversion of contaminants to H₂O and CO₂). However, no results related to mineralization are found in the manuscript. Therefore, the authors should provide total organic carbon (TOC) removal data.

Response: We appreciate the reviewer's insightful comment, and in response, we have conducted additional analyses to measure the total organic carbon (TOC) values before and after the reaction (shown as Figure R5 below). The observed 72.0% reduction in TOC after the degradation test indicates the robust mineralization capability inherent in our reaction system. Figure R5 has been added as Figure S8b of the revised manuscript, with a proper description.

Figure R5. Comparison of the TOC values before and after a 2-hour degradation of BPA. Conditions: catalyst: 0.15 g/L, BPA: 0.1 mM, PDS: 1 mM.

2) The performance of the integrated modular device needs to be validated in real wastewater. Once this is addressed, the practicality of this device will be more apparent.

Response: We appreciate the insightful comment from the reviewer. To explore the performance of our device towards real surface water, we took water sample directly from the Yuhangtang River, located at Zhejiang University's Zijiang campus in Hangzhou, Zhejiang province, China (Figure R6a below). The collected river water, augmented with a BPA solution, was utilized to directly evaluate the treatment efficacy employing our floating device (Figure R6b). As depicted in Figure R6c, it showed an obvious removal of BPA in the real river water.

Figure R6 has been added into the revised manuscript as Figure S30, with a proper description.

Figure R6. (a) Photographs of the river water and deionized water samples. (b) Photograph of the floating device set-up placed on the water tank with 2 L of BPA-contaminated river water. (c) Monitoring of the change in the corresponding BPA concentration within the tank. Conditions: BPA, 5 μ M, 2 L; PDS, 10 mM, 40 mL.

REVIEWERS' COMMENTS

Reviewer #1 (Remarks to the Author):

The authors have made the necessary level of additional evidence or rebuttals to the comments raised by the reviewers. The main concern is the applicability and cost prohibitiveness of the devices used for the "purification" of large scale water body. The authors have acknowledged that this is a proof-of-concept study and may have other applications beyond surface water treatment.

Reviewer #2 (Remarks to the Author):

This reviewer does not have any additional comments.

Reviewer #3 (Remarks to the Author):

This reviewer has no further comments and questions. This paper has been well revised and is acceptable in its current form.

Responses to the reviewers' comments (Manuscript number NCOMMS-23-14745B)

Changes in the revised manuscript as a response to the reviewers' comments are highlighted in red color and clarifications regarding the reviewer's comments are provided in blue color.

Reviewer #1 (Remarks to the Author):

The authors have made the necessary level of additional evidence or rebuttals to the comments raised by the reviewers. The main concern is the applicability and cost prohibitiveness of the devices used for the "purification" of large scale water body. The authors have acknowledged that this is a proof-of-concept study and may have other applications beyond surface water treatment.

Response: We appreciate the thoughtful review and the recognition of our research's contributions toward demonstrating a new strategy for potential large scale water treatment. The reviewer's insights have been invaluable in shaping our work. For the use of this strategy for real application in large scale water body, we are working on reduction of the price and optimization of the device structure, and hope we can present more interesting work in the near future.

Reviewer #2 (Remarks to the Author):

This reviewer does not have any additional comments.

Response: We appreciate the thoughtful review on our work. The reviewer's insights have been invaluable in shaping our work.

Reviewer #3 (Remarks to the Author):

This reviewer has no further comments and questions. This paper has been well revised and is acceptable in its current form.

Response: We appreciate the thoughtful review on our work. The reviewer's insights have been invaluable in shaping our work.